

# INSTRUCTOR: Instructing Unsupervised Conversational Dense Retrieval with Large Language Models

**Zhuoran Jin**[1,2], **Pengfei Cao**[1,2], **Yubo Chen**[1,2,*], **Kang Liu**[1,2], **Jun Zhao**[1,2]

[1] School of Artificial Intelligence, University of Chinese Academy of Sciences, Beijing, China
[2] The Laboratory of Cognition and Decision Intelligence for Complex Systems,
Institute of Automation, Chinese Academy of Sciences, Beijing, China
{zhuoran.jin, pengfei.cao, yubo.chen, kliu, jzhao}@nlpr.ia.ac.cn

## Abstract

Compared to traditional single-turn ad-hoc retrieval, conversational retrieval needs to handle the multi-turn conversation and understand the user's real query intent. However, most existing methods simply fine-tune the pre-trained ad-hoc retriever on limited supervised data, making it challenging for the retriever to fully grasp the entirety of the conversation. In this paper, we find that large language models (LLMs) can accurately discover the user's query intent from the complex conversation context and provide the supervised signal to instruct the retriever in an *unsupervised* manner. Therefore, we propose a novel method termed 🦙 INSTRUCTOR to **Instruct** unsupervised c**O**nversational dense **R**etrieval with LLMs. We design an unsupervised training framework that employs LLMs to estimate the session-passage relevance score as the soft label to guide the retriever's training. Specially, we devise three instructing strategies from *context*, *query* and *response* perspectives to calculate the relevance score more precisely, including conversational retrieval as conversation generation, question rewrite as latent variable and question response as posterior guide. Experimental results show INSTRUCTOR can bring significant improvements across various ad-hoc retrievers, even surpassing the current supervised state-of-the-art method. We also demonstrate the effectiveness of our method under low-resource and zero-shot settings. Our code is publicly available at GitHub [1].

## 1 Introduction

The development of generative language models ([Brown et al., 2020](#); [OpenAI, 2023](#)) has sparked a paradigm shift in information-seeking, transitioning from using search engines to interacting with conversational assistants (*e.g.*, ChatGPT). Conversational assistants offer a more flexible and user-friendly experience by addressing users' queries

through multi-turn conversations. However, they still struggle to provide highly *accurate* and *up-to-date* responses. To mitigate the problems, one promising solution ([Izacard and Grave, 2021](#)) is to employ a *retrieve-then-generate* pipeline. As illustrated in Figure [1](#)(a), to answer the user's conversational query "*Did he like collecting things?*" in the conversation, the pipeline first retrieves a small set of reference passages about "*Nicolas-Claude Fabri de Peiresc*" via **conversational retrieval** and then generates the response conditioned on these passages along with the ongoing conversation.

Conversational retrieval, which focuses on retrieving relevant passages from a large passage collection based on the given conversation session, has gained considerable attention from researchers. Unlike traditional **single-turn** ad-hoc retrieval, conversational retrieval needs to deal with the complex **multi-turn** conversation session which consists of both the historical conversation context and the user's current query. In contrast to standalone queries, conversational queries may contain more complicated linguistic phenomena, such as *omissions*, *coreferences*, and *ambiguities* shown in Figure [1](#)(a). Besides, *topic-switching* behaviour is natural in information-seeking conversations ([Adlakha et al., 2022](#)), introducing a significant amount of noise in the conversation context that is unrelated to the user's query. Therefore, conversational retrieval is much more challenging than ad-hoc retrieval.

Existing methods can be mainly categorized into conversational query rewriting and conversational dense retrieval. Conversational query rewriting methods ([Yu et al., 2020](#); [Lin et al., 2020](#)) utilize the rewriting models to explicitly reformulate the conversational queries into de-contextualized rewrites and then perform ad-hoc retrieval. While such a two-stage approach offers strong interpretability, its rewriting stage poses difficulties in direct optimization for retrieval performance and leads to increased retrieval latency ([Mao et al., 2022a](#)).

---

[1] https://github.com/jinzhuoran/InstructoR/
[*] Corresponding author.

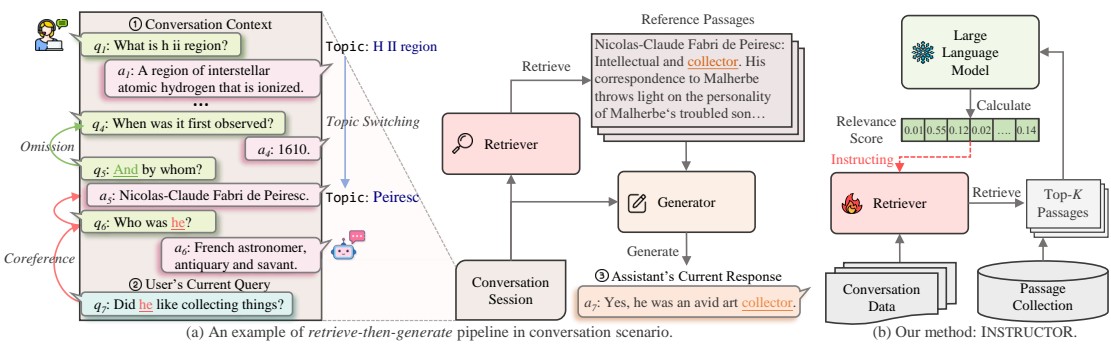

(a) An example of *retrieve-then-generate* pipeline in conversation scenario.    (b) Our method: INSTRUCTOR.

Figure 1: The illustration of a *retrieve-then-generate* pipeline and our proposed *unsupervised* method.

Recently, conversational dense retrieval methods (Lin et al., 2021; Mao et al., 2023b) have shown better retrieval effectiveness. Most existing supervised methods leverage the pre-trained ad-hoc retriever to encode the entire conversation session into a dense embedding and then fine-tune the retriever on conversational retrieval data. However, they still face the following two challenges: (1) Fine-tuning the retriever often requires a large number of labeled session-passage pairs. In practical applications, **annotating session-passage pairs is much more difficult than collecting conversation data**. Hence, there is a need to train a conversational dense retriever in an *unsupervised* manner, *i.e.*, without using session-passage pairs; (2) Simply compressing the session into one single vector may mix irrelevant context and neglect crucial information, **making it hard for the retriever to comprehend the user's query in the complex conversation**.

In this paper, we propose a novel method termed 🧑‍🏫 INSTRUCTOR to **Instruct** unsupervised c**O**nversational dense **R**etrieval with large language models (LLMs). The main insight of our INSTRUCTOR is that the powerful LLMs can help us train ad-hoc retrievers for conversational retrieval without supervision. On the one hand, some research (Liu et al., 2022; Meng et al., 2022; Dai et al., 2022) has demonstrated that LLMs can generate high-quality annotated data with only a few or even without demonstration examples. On the other hand, we find that LLMs have powerful linguistic capabilities (Liang et al., 2022) and can accurately grasp the user's intent from the complex and noisy conversation context. Based on these findings, we unleash the power of LLMs to judge the relevance between sessions and passages without requiring any training. Specifically, we design a *three-stage* unsupervised training framework for INSTRUCTOR shown in Figure 1(b). Given a session, INSTRUCTOR first

retrieves top-$K$ relevant passages via ad-hoc retrievers, then generates session-passage **relevance score** with frozen LLMs to rerank the top-$K$ passages, and finally uses relevance scores to guide the iterative training of conversational retrieval.

The core of INSTRUCTOR lies in how to utilize LLMs to calculate relevance scores. Since the generative process requires LLMs to focus on every token in the session and passage, we approximate the relevance score based on the LLM's conditional generation log likelihood. To obtain a more accurate score, we propose three instructing strategies from the perspectives of *context*, *query*, and *response* within the conversation, as outlined below: (1) **Conversational Retrieval as Conversation Generation**: To avoid LLMs being distracted by lengthy and irrelevant *context*, we decouple conversational dense retrieval into instructional generation subtasks, including conditional context generation and question generation; (2) **Question Rewrite as Latent Variable**: To resolve the user's ambiguous *query*, we model question rewrites generated by black-box LLMs as latent variables to implicitly guide the retriever training, showing superior effectiveness and efficiency compared to the explicit usage of question rewrites for retrieval; (3) **Question Response as Posterior Guide**: To capture relevance precisely, we find that question *response* can provide the relevant signal when the passages related to the session are unknown. Therefore, we treat question responses as the posterior guide to further enhance the retriever. We conduct extensive experiments on four datasets and prove all three strategies can bring significant improvements across various ad-hoc retrievers. Moreover, our method surpasses the current supervised state-of-the-art model without using labeled training data.

Our contributions are summarized as follows:

- We propose a novel method called INSTRUC-

TOR to instruct unsupervised conversational dense retrieval with LLMs. To the best of our knowledge, this is the first attempt to utilize LLMs to empower conversational retrieval.

- We devise three instructing strategies to calculate session-passage relevance score, including conversational retrieval as conversation generation, question rewrite as latent variable and question response as posterior guide.

- We demonstrate INSTRUCTOR can bring consistent and significant improvements across various ad-hoc retrievers, with an average Recall@100 improvement of 9.0% and 34.1% on QReCC and TopiOCQA, even surpassing the current supervised state-of-the-art method by 3.2% and 9.7%. We also evaluate our method under low-resource and zero-shot settings.

## 2 Preliminary

### 2.1 Task Definition

We formulate the conversational retrieval task as finding the relevant passage $d$ from a large passage collection $\mathcal{D}$ based on the given conversation session $s_t = \{c_{t-1}, q_t\}$, where $c_{t-1} = \{q_1, a_1, q_2, a_2, ..., q_{t-1}, a_{t-1}\}$ denotes the historical conversation context consisting of $t-1$ question-response pairs $(q, a)$, and $q_t$ denotes the user's query of $t$-th turn. Considering the length limitation of the retriever's input, we represent the input format of the session $s_t$ as:

$$s_t = [\text{CLS}] \oplus q_t \oplus a_{t-1} \oplus \cdots \oplus a_1 \oplus q_1 \oplus [\text{SEP}], \quad (1)$$

where $[\text{CLS}]$ and $[\text{SEP}]$ are special tokens, $\oplus$ denotes concatenation. We demonstrate that this reverse concatenation works well in Appendix A. We denote the query rewrite of $q_t$ as $r_t$. For simplicity, we omit the subscript $t$ in the rest of this paper.

### 2.2 Embedding Similarity as Retrieval Score

We adopt the pre-trained ad-hoc retriever as the strong initial conversational retriever, which uses a dual-encoder architecture where sessions and passages are encoded independently. Given a session $s$ and a passage $d$, we encode them with session encoder $f_s$ and passage encoder $f_d$ respectively. The retrieval score $\text{sim}(s, d)$ between a session $s$ and a passage $d$ is the dot product of the embeddings:

$$\text{sim}(s, d) = \langle f_s(s), f_d(d) \rangle. \quad (2)$$

Despite the dual-encoder's efficiency, the limited model parameters and embedding dimensions hinder its ability to comprehend sessions and passages.

### 2.3 LLM Likelihood as Relevance Score

Given a text pair $(s, d)$, we employ generative LLMs to compute the relevance score $\text{score}(d \mid s)$ without any training examples, which approximates the probability of retrieving $d$ based on $s$. Since the generative process requires LLMs to focus on and explain every token in $s$ and $d$, we estimate the relevance score $\text{score}(d \mid s)$ as the generation likelihood of $d$ conditioned on $s$:

$$\begin{aligned} \text{score}(d \mid s) &\approx p(d \mid s) \propto \log p(d \mid s) \\ &\propto \frac{1}{m} \sum_{k=1}^{m} \log p(d_k \mid I, s, d_{<k}), \end{aligned} \quad (3)$$

where $I$ is the task-specific generation instruction, and $m$ is the sequence length of $d$. We collectively refer to those resource-intensive LLMs (>100B) which allow limited access through commercial APIs, as *black-box* LLMs (*e.g.*, ChatGPT and GPT-4). Conversely, we refer to those cost-effective LLMs (<10B) that can be called millions of times during training to calculate $\text{score}(d \mid s)$, as *white-box* LLMs (*e.g.*, Flan-T5 (Chung et al., 2022) and GPT-Neo (Andonian et al., 2021)).

## 3 Methodology: INSTRUCTOR

In this paper, we consider a more realistic setting which is to train a conversational dense retriever in an *unsupervised* manner, *i.e.*, without using session-passage pairs. We only have a certain number of conversation sessions and a large passage collection during the training. Our INSTRUCTOR uses the frozen LLMs' conditional generation log likelihood as the supervised signal to guide the training of the retriever without direct supervision. According to Sachan et al. (2022b), we adopt an unsupervised training framework to distill knowledge from LLM to retriever. Moreover, we devise three instructing strategies from diverse perspectives in the conversation to calculate the relevance score of session-passage pairs more accurately.

### 3.1 Unsupervised Training Framework

In the unsupervised training framework of INSTRUCTOR, we adopt a three-stage approach to instruct the ad-hoc retriever with LLM as shown in Figure 2. For fast passage retrieval, we apply the passage encoder $f_d$ to the passage collection $\mathcal{D}$ and build passage index $\mathcal{I}$. We froze $f_d$ to avoid rebuilding the index during the training.

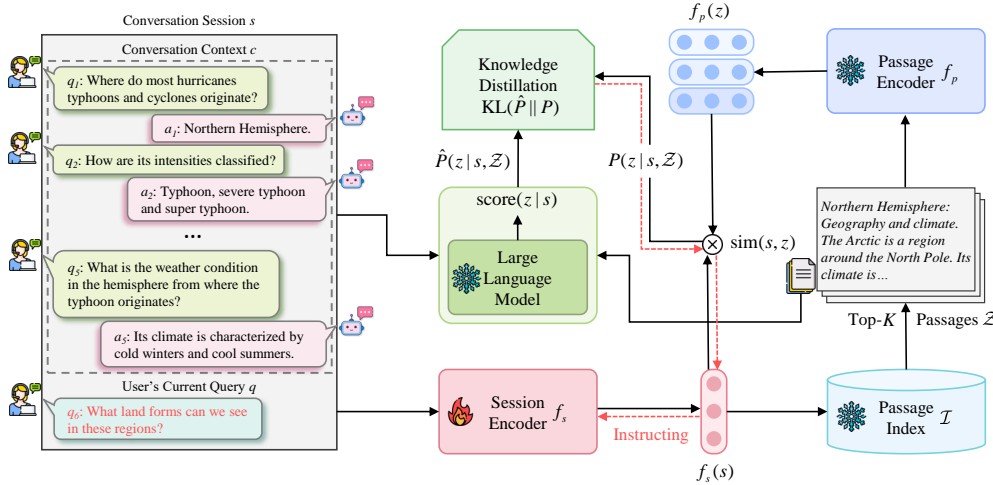

Figure 2: The unsupervised training framework of INSTRUCTOR. Red dotted arrows indicate the gradient flow.

**Compute Top-$K$ Retrieval Probability.** Given a conversation session $s$, we first retrieve the top-$K$ passages $\mathcal{Z}$ with the retriever (*student*) and compute the retrieval probability of passage $z \in \mathcal{Z}$:

$$P\left(z \mid s, \mathcal{Z}\right) = \frac{\exp\left(\text{sim}\left(s, z\right)/\tau\right)}{\sum_{z' \in \mathcal{Z}} \exp\left(\text{sim}\left(s, z'\right)/\tau\right)}, \quad (4)$$

where $\tau$ denotes a temperature hyperparameter.

**Generate Top-$K$ Relevance Probability.** Since there are no supervised labels for sessions and passages, we use the LLM (*instructor*) to generate the relevance probability of passage $z$ conditioned on session $s$ as the soft supervised signal:

$$\hat{P}\left(z \mid s, \mathcal{Z}\right) = \frac{\exp\left(\text{score}\left(z \mid s\right)\right)}{\sum_{z' \in \mathcal{Z}} \exp\left(\text{score}\left(z' \mid s\right)\right)}. \quad (5)$$

We provide three different strategies to derive the score$(z \mid s)$ formula in Section 3.2.

**Distill Knowledge from LLM to Retriever.** We train the retriever by minimizing the KL divergence between the relevance probability $\hat{P}\left(z \mid s, \mathcal{Z}\right)$ and the retrieval probability $P\left(z \mid s, \mathcal{Z}\right)$:

$$\mathcal{L} = \sum_{z \in \mathcal{Z}} \hat{P}\left(z \mid s, \mathcal{Z}\right) \log \frac{\hat{P}\left(z \mid s, \mathcal{Z}\right)}{P\left(z \mid s, \mathcal{Z}\right)}. \quad (6)$$

We only update the parameters of session encoder $f_s$ during the training.

## 3.2 Instructing Strategies

In this section, we measure the relevance score of retrieving passage $z$ according to the session $s$ via score$(z \mid s)$, which can be estimated as the LLM's generation log likelihood $p(z \mid s)$ of $z$ conditioned on $s$. To calculate the relevance score more precisely, we will detail our proposed three instructing strategies from *context*, *query* and *response* perspectives within the conversation.

**Conversational Retrieval as Conversation Generation (INSTRUCTOR_CRCG).** We find that if we calculate $p(z \mid s)$ by directly generating $z$ conditioned on $s$ with white-box LLMs, it causes LLMs to be distracted by the lengthy and noisy conversation context $c$, and unable to focus on the current query $q$. Therefore, we reformulate conversational retrieval as a task of generating conversation session $s$ conditioned on given passage $z$, which can be further decoupled into two instructional generation subtasks based on Bayes' Theorem:

$$
\begin{aligned}
\text{score}_c(z \mid s) &\approx p(z \mid s) \propto \log p(z \mid c, q) \\
&= \log \frac{p(z, c, q)}{p(c, q)} \\
&= \log \frac{p(z)p(c \mid z)p(q \mid z, c)}{p(c)p(q \mid c)} \\
&\propto \log p(c \mid I_z^c, z) + \log p(q \mid I_{z,c}^q, z, c),
\end{aligned}
\quad (7)
$$

where $\log p(z)$, $\log p(c)$ and $\log p(q \mid c)$ are assumed as constants and can be ignored. As shown in Figure 3(a), we first prompt LLMs with an instruction $I_z^c$ "*Please create a conversation:*" to generate the conversation context $c$ condition on the retrieved passage $z$, then prompt LLMs with another instruction $I_{z,c}^q$ "*Please continue to write a question:*" to generate the current question $q$ condition on the passage $z$ and the ongoing conversation context $c$. Hence, we can combine $\log p(c \mid I_z^c, z)$ and $\log p(q \mid I_{z,c}^q, z, c)$ together as the estimate of relevance score score$_c(z \mid s)$ between $s$ and $z$.

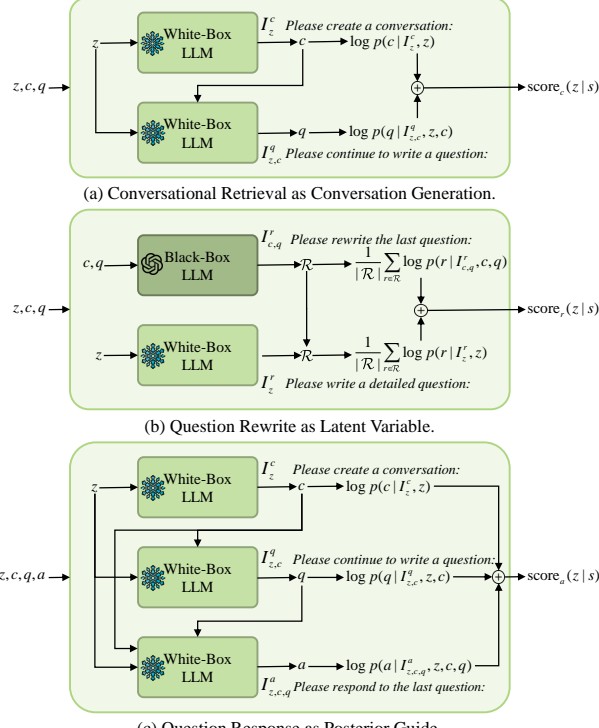

(a) Conversational Retrieval as Conversation Generation.

(b) Question Rewrite as Latent Variable.

(c) Question Response as Posterior Guide.

Figure 3: Three instructing strategies of INSTRUCTOR.

**Question Rewrite as Latent Variable (INSTRUCTOR_QRLV).** We reveal that conversational query rewriting exhibits outstanding performance in topic-switching scenarios illustrated in Appendix A. To solve conversational retrieval from the query perspective, we utilize both black-box and white-box LLMs to rewrite question $q$. As shown in Table 4, we find that black-box LLMs have a much higher rewrite quality than white-box LLMs, indicating that black-box LLMs have more powerful linguistic capabilities to understand the conversations. However, suffering from limited API usage, we cannot completely use black-box LLMs in INSTRUCTOR_CRCG. To augment the retriever with the ability of black-box LLMs to tackle the user's ambiguous query $q$, we model a query rewrite $r$ as a latent variable and marginalizing over all possible rewrites $\mathcal{R}$:

$$
\begin{aligned}
\text{score}_r(z \mid s) &\approx p(z \mid s) \propto \log p(z \mid r)p(r \mid c, q) \\
&= \log \frac{p(r \mid z)p(z)p(r \mid c, q)}{p(r)} \\
&\propto \log p(r \mid I_{c,q}^r, c, q) + \log p(r \mid I_z^r, z) \\
&\propto \frac{1}{|\mathcal{R}|} \sum_{r \in \mathcal{R}} \left( \log p(r \mid I_{c,q}^r, c, q) + \log p(r \mid I_z^r, z) \right).
\end{aligned}
\tag{8}
$$

As shown in Figure 3(b), we first instruct black-box LLMs to generate a set of potential rewrites $r \in \mathcal{R}$ with confidence $\log p(r \mid I_{c,q}^r, c, q)$ based on the

given session $s$. Then we treat $R$ as the references for white-box LLMs to compute $\log p(r \mid I_z^r, z)$ during the training. In practical implementation, we pre-generate the rewrites for all training sessions, thereby avoiding redundant calls to black-box LLMs. Our strategy greatly reduces the cost of employing black-box LLMs and demonstrates superior effectiveness and efficiency compared to the explicit usage of question rewrites for retrieval.

**Question Response as Posterior Guide (INSTRUCTOR_QRPG).** We find that $p(z \mid s, a)$ can serve as a more potent supervised signal than $p(z \mid s)$, as demonstrated in Appendix A. Here, $a$ refers to the gold response or is generated by black-box language models. Therefore, we argue that question response $a$ can provide additional relevance judgement when the passages related to the question are unknown under the unsupervised setting. To precisely identify relevant passages, we incorporate question response generation task into INSTRUCTOR_CRCG as the posterior guide:

$$
\begin{aligned}
\text{score}_a(z \mid s) &\approx p(z \mid s) \propto \log p(z \mid c, q, a) \\
&= \log \frac{p(z, c, q, a)}{p(c, q, a)} \\
&= \log \frac{p(z)p(c \mid z)p(q \mid z, c)p(a \mid z, c, q)}{p(c)p(q \mid c)p(a \mid c, q)} \\
&\propto \log p(c \mid I_z^c, z) + \log p(q \mid I_{z,c}^q, z, c) \\
&+ \log p(a \mid I_{z,c,q}^a, z, c, q),
\end{aligned}
\tag{9}
$$

where $\log p(a \mid c, q)$ is irrelevant to $z$, so it is eliminated. As shown in Figure 3(c), we prompt LLMs with instruction $I_{z,c,q}^a$ "*Please respond to the last question:*" to generate the response $a$ for the last question $q$, considering the reference passage $z$ and the conversation session $s$. $\log p(a \mid I_{z,c,q}^a, z, c, q)$ indicates that more informative passage $z$ can assist LLMs in generating response $a$ more effortlessly.

## 4 Experimental Setup

### 4.1 Datasets and Evaluation

**Datasets.** We evaluate our INSTRUCTOR in three experimental settings. The first is the *unsupervised* setting conducted on two popular conversational retrieval datasets: QReCC (Anantha et al., 2021) and TopiOCQA (Adlakha et al., 2022). QReCC is a large-scale dataset for open-domain conversational question answering (ODCQA) with human-rewritten query rewrites. TopiOCQA is a more challenging dataset for ODCQA under topic-switching

| Retriever | QReCC | | | | TopiOCQA | | | |
|---|---|---|---|---|---|---|---|---|
| | **MRR** | **NDCG@3** | **R@10** | **R@100** | **MRR** | **NDCG@3** | **R@10** | **R@100** |
| *Unsupervised Approaches* | | | | | | | | |
| Rewrite$_{Contriever}$ | 19.7 | 17.0 | 33.6 | 56.4 | 12.4 | 10.3 | 26.2 | 52.4 |
| Rewrite$_{Contriever-msmarco}$ | 33.8 | 30.9 | 52.5 | 70.4 | 30.0 | 28.3 | 51.3 | 72.3 |
| HyDE$_{Contriever}$ | 40.4 | 37.3 | 63.0 | 82.9 | 24.1 | 22.0 | 45.2 | 69.8 |
| HyDE$_{Contriever-msmarco}$ | 43.3 | 40.5 | 65.4 | 84.6 | 35.3 | 34.1 | 59.0 | 77.8 |
| query2doc$_{Contriever}$ | 7.4 | 6.5 | 12.5 | 19.1 | 12.2 | 9.9 | 24.4 | 52.1 |
| query2doc$_{Contriever-msmarco}$ | 29.4 | 27.1 | 45.3 | 62.8 | 25.3 | 23.2 | 45.9 | 71.3 |
| DPR | 28.7 | 26.3 | 43.0 | 60.7 | 10.7 | 9.8 | 18.2 | 33.0 |
| w/ INSTRUCTOR$_{CRCG}$ | 34.5 $\uparrow^{5.8}$ | 31.8 $\uparrow^{5.5}$ | 55.1 $\uparrow^{12.1}$ | 73.5 $\uparrow^{12.8}$ | 20.2 $\uparrow^{9.5}$ | 18.8 $\uparrow^{9.0}$ | 35.4 $\uparrow^{17.2}$ | 59.7 $\uparrow^{26.7}$ |
| w/ INSTRUCTOR$_{QRLV}$ | 30.1 $\uparrow^{1.4}$ | 26.4 $\uparrow^{0.1}$ | 52.3 $\uparrow^{9.3}$ | 75.4 $\uparrow^{14.7}$ | 21.8 $\uparrow^{11.1}$ | 20.1 $\uparrow^{10.3}$ | 39.8 $\uparrow^{21.6}$ | 64.5 $\uparrow^{31.5}$ |
| w/ INSTRUCTOR$_{QRPG}$ | 36.2 $\uparrow^{7.5}$ | 33.3 $\uparrow^{7.0}$ | 57.0 $\uparrow^{14.0}$ | 75.6 $\uparrow^{14.9}$ | 29.9 $\uparrow^{19.2}$ | 28.3 $\uparrow^{18.5}$ | 50.3 $\uparrow^{32.1}$ | 72.6 $\uparrow^{39.6}$ |
| ANCE | 41.4 | 38.9 | 61.2 | 74.7 | 11.6 | 10.2 | 21.8 | 40.1 |
| w/ INSTRUCTOR$_{CRCG}$ | 42.0 $\uparrow^{0.6}$ | 39.1 $\uparrow^{0.2}$ | 64.6 $\uparrow^{3.4}$ | 83.0 $\uparrow^{8.3}$ | 18.2 $\uparrow^{6.6}$ | 16.8 $\uparrow^{6.6}$ | 33.3 $\uparrow^{11.5}$ | 56.4 $\uparrow^{16.3}$ |
| w/ INSTRUCTOR$_{QRLV}$ | 42.7 $\uparrow^{1.3}$ | 40.2 $\uparrow^{1.3}$ | 63.4 $\uparrow^{2.2}$ | 78.4 $\uparrow^{3.7}$ | 17.9 $\uparrow^{6.3}$ | 15.8 $\uparrow^{5.6}$ | 34.8 $\uparrow^{13.0}$ | 59.7 $\uparrow^{19.6}$ |
| w/ INSTRUCTOR$_{QRPG}$ | 43.5 $\uparrow^{2.1}$ | 40.5 $\uparrow^{1.6}$ | 66.7 $\uparrow^{5.5}$ | 85.6 $\uparrow^{10.9}$ | 25.3 $\uparrow^{13.7}$ | 23.7 $\uparrow^{13.5}$ | 45.1 $\uparrow^{23.3}$ | 69.0 $\uparrow^{28.9}$ |
| Contriever | 41.5 | 38.7 | 63.8 | 84.3 | 6.9 | 5.3 | 14.5 | 37.5 |
| w/ INSTRUCTOR$_{CRCG}$ | 48.7 $\uparrow^{7.2}$ | 45.9 $\uparrow^{7.2}$ | 73.8 $\uparrow^{10.0}$ | 90.9 $\uparrow^{6.6}$ | 16.4 $\uparrow^{9.5}$ | 13.7 $\uparrow^{8.4}$ | 34.0 $\uparrow^{19.5}$ | 67.5 $\uparrow^{30.0}$ |
| w/ INSTRUCTOR$_{QRLV}$ | 43.6 $\uparrow^{2.1}$ | 40.4 $\uparrow^{1.7}$ | 69.0 $\uparrow^{5.2}$ | 90.7 $\uparrow^{6.4}$ | 18.6 $\uparrow^{11.7}$ | 15.9 $\uparrow^{10.6}$ | 37.7 $\uparrow^{23.2}$ | 71.8 $\uparrow^{34.3}$ |
| w/ INSTRUCTOR$_{QRPG}$ | 50.0 $\uparrow^{8.5}$ | 47.2 $\uparrow^{8.5}$ | 74.5 $\uparrow^{10.7}$ | 90.4 $\uparrow^{6.1}$ | 27.4 $\uparrow^{20.5}$ | 25.2 $\uparrow^{19.9}$ | 51.3 $\uparrow^{36.8}$ | 78.1 $\uparrow^{40.6}$ |
| Contriever-msmarco | 47.6 | 45.0 | 70.5 | 88.9 | 16.2 | 14.3 | 29.4 | 56.0 |
| w/ INSTRUCTOR$_{CRCG}$ | 50.0 $\uparrow^{2.4}$ | 47.2 $\uparrow^{2.2}$ | 74.9 $\uparrow^{4.4}$ | 92.1 $\uparrow^{3.2}$ | 23.9 $\uparrow^{7.7}$ | 21.6 $\uparrow^{7.3}$ | 44.4 $\uparrow^{15.0}$ | 73.7 $\uparrow^{17.7}$ |
| w/ INSTRUCTOR$_{QRLV}$ | 44.1 $\downarrow^{3.5}$ | 40.7 $\downarrow^{4.3}$ | 70.6 $\uparrow^{0.1}$ | 92.0 $\uparrow^{3.1}$ | 28.6 $\uparrow^{12.4}$ | 26.0 $\uparrow^{11.7}$ | 52.3 $\uparrow^{22.9}$ | 80.2 $\uparrow^{24.2}$ |
| w/ INSTRUCTOR$_{QRPG}$ | **52.9** $\uparrow^{5.3}$ | **50.4** $\uparrow^{5.4}$ | **77.7** $\uparrow^{7.2}$ | **92.9** $\uparrow^{4.0}$ | **38.5** $\uparrow^{22.3}$ | **37.0** $\uparrow^{22.7}$ | **62.1** $\uparrow^{32.7}$ | **83.2** $\uparrow^{27.2}$ |
| *Supervised Approaches* | | | | | | | | |
| DPR$^{FT}$ | 35.0 | 32.4 | 54.9 | 75.2 | 31.6 | 29.8 | 53.3 | 76.3 |
| ANCE$^{FT}$ | 48.2 | 45.3 | 70.6 | 88.3 | 32.6 | 31.3 | 53.7 | 73.3 |
| Contriever$^{FT}$ | 48.7 | 46.0 | 74.7 | 92.1 | 34.4 | 32.9 | 56.5 | 78.5 |
| Contriever-msmarco$^{FT}$ | 50.9 | 48.1 | 76.0 | 92.2 | 35.0 | 33.6 | 58.3 | 78.9 |
| SPLADE$^{FT\dagger}$ | 50.0 | 46.6 | 69.9 | 87.8 | 30.7 | 29.5 | 52.1 | 72.0 |
| ConvDR$^{\dagger}$ | 38.5 | 35.7 | 58.2 | 77.8 | 27.2 | 26.4 | 43.5 | 61.1 |
| LeCoRE$^{\dagger}$ | 51.1 | 48.5 | 73.9 | 89.7 | 32.0 | 31.4 | 54.3 | 73.5 |

Table 1: Experimental results of unsupervised setting on QReCC and TopiOCQA. The best results are highlighted in **bold**. † indicates the results taken from corresponding original papers.

scenarios. The second is the *low-resource* setting, we sample different numbers of conversation data from TopiOCQA. The third is the *zero-shot* setting, we train the models on the TopiOCQA training set, then directly evaluate them on two conversational search test sets: CAsT-19 (Dalton et al., 2020) and CAsT-20 (Dalton et al., 2021). More details about these four datasets are shown in Appendix C.

**Evaluation Metrics.** In line with prior studies (Mao et al., 2023b), we employ four metrics (%): MRR, NDCG@3, Recall@10, and Recall@100 to comprehensively evaluate the retrieval performance. We use pytrec_eval toolkit (Van Gysel and de Rijke, 2018) to calculate these metrics.

### 4.2 Implementation Details

**Ad-hoc Retrievers.** We adopt four representative ad-hoc retrievers as initial retrievers in INSTRUCTOR: (1) DPR (Karpukhin et al., 2020): A retriever trained on Natural Questions (Kwiatkowski et al., 2019); (2) ANCE (Xiong et al., 2021): A retriever

trained on MS MARCO (Nguyen et al., 2016); (3) Contriever (Izacard et al., 2022): An unsupervised retriever trained with contrastive learning; (4) Contriever-msmarco: Contriever fine-tuned on MS MARCO. Except ANCE is initialized with RoBERTa (Liu et al., 2019), all other retrievers are initialized with BERT (Devlin et al., 2019).

**Large Language Models.** We adopt Flan-T5 XL (3B) as the white-box LLM and gpt-3.5-turbo as the black-box LLM. More details about hyperparameter settings can be found in Appendix E.

### 4.3 Baselines

**Unsupervised Methods.** We compare INSTRUCTOR with three unsupervised methods that also adopt LLMs: (1) Rewrite: We use an LLM to reformulate the query, then perform ad-hoc retrieval; (2) HyDE (Gao et al., 2022): An unsupervised method without training the retriever, which adopts an LLM to generate hypothetical documents for the query, then retrieves real documents with hypothetical

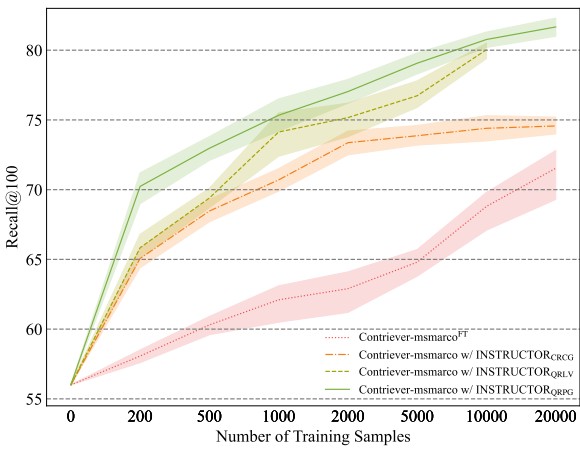

Figure 4: Experimental results of low-resource setting on TopiOCQA.

documents; (3) query2doc (Wang et al., 2023): A query expansion approach, which expands the original query with LLM's generated documents. For a fair comparison, we all adopt `gpt-3.5-turbo` as the LLM, Contriever and Contriever-msmarco as the ad-hoc retrievers to reproduce these three methods on the conversational retrieval task.

**Supervised Methods.** We also compare our INSTRUCTOR with several supervised methods: (1) FT: We fine-tune DPR, ANCE, Contriever and Contriever-msmarco on supervised data; (2) SPLADE[FT] (Formal et al., 2021): A sparse lexical-based retriever fine-tuned on supervised data; (3) ConvDR (Yu et al., 2021): ANCE fine-tuned on supervised data using knowledge distillation between the query rewrite representation and the latent session representation; (4) LeCoRE (Mao et al., 2023b): The current supervised state-of-the-art method, which extends SPLADE with two well-matched multi-level denoising methods.

## 5 Experimental Results

### 5.1 Unsupervised Conversational Retrieval

The experimental results of the unsupervised setting on QReCC and TopiOCQA are presented in Table 1, where we have the following observations:

(1) INSTRUCTOR can bring consistent and significant improvements across various ad-hoc retrievers in Recall@100, with an average improvement of 9.0% on QReCC and 34.1% on TopiOCQA. Remarkable improvement on TopiOCQA shows that our method is effective under challenging topic-switching scenarios, indicating that LLMs can help ad-hoc retrievers understand the complex conversa-

tion session and discover the user's query intent.

(2) Compared to other unsupervised methods that employ LLMs to generate query rewrites or hypothetical documents during inference time, our trainable method achieves superior performance. We argue that using LLMs to dynamically guide the training of retrievers enables a better transition from ad-hoc retrieval to conversational retrieval.

(3) Without using any labeled training data, IN-STRUCTOR surpasses the current supervised state-of-the-art method LeCoRE, with Recall@100 improvements of 3.2% and 9.7% on QReCC and TopiOCQA. Furthermore, our method can achieve comparable or even better performance than direct fine-tuning on supervised data. This proves that LLMs have the *zero-shot* ability to measure text relevance.

(4) Experimental results validate the effectiveness of all three proposed instructing strategies. IN-STRUCTOR[CRCG], which solely relies on white-box LLMs, gains consistent improvements on QReCC and TopiOCQA. INSTRUCTOR[QRLV], which incorporates more powerful black-box LLMs, improves significantly on TopiOCQA, but not so much on QReCC. These findings align with our insights presented in Appendix A. Notably, INSTRUCTOR[QRPG] achieves the best retrieval performance, demonstrating the efficacy of utilizing question responses as posterior guidance for the retrievers.

### 5.2 Low-Resource Conversational Retrieval

We perform an evaluation of low-resource conversational retrieval on TopiOCQA, using different numbers of conversation data for training to simulate the low-resource scenario. As illustrated in Figure 4, our method can better utilize a small number of training samples than directly fine-tuning the retriever. Only needing 200 samples can bring a 10%-15% performance improvement to the ad-hoc retriever. Besides, INSTRUCTOR[QRLV] is data-efficient, requiring only 10,000 training samples (costing $5) to achieve excellent performance.

### 5.3 Zero-Shot Conversational Retrieval

We also conduct zero-shot conversational retrieval to evaluate the transferability. We first train the retrievers on the TopiOCQA training set and then directly evaluate them on CAsT test sets. According to Table 3, we note the following key findings:

(1) INSTRUCTOR exhibits an average MRR improvement of 14.5% on CAsT-19 and 18.3% on CAsT-20, highlighting its strong generalization.

| Retriever | CAsT-19 | | | | CAsT-20 | | | |
|---|---|---|---|---|---|---|---|---|
| | MRR | NDCG@3 | R@10 | R@100 | MRR | NDCG@3 | R@10 | R@100 |
| *w/o Fine-tuning* | | | | | | | | |
| Rewrite$_{Contriever}$ | 40.6 | 24.5 | 6.4 | 26.1 | 21.7 | 12.0 | 7.5 | 25.1 |
| Rewrite$_{Contriever-msmarco}$ | 63.6 | 48.2 | 12.2 | 43.8 | 39.6 | 25.7 | 14.6 | 43.5 |
| HyDE$_{Contriever}$ | 54.7 | 38.9 | 9.3 | 33.2 | 44.8 | 33.2 | 16.0 | 44.4 |
| HyDE$_{Contriever-msmarco}$ | 61.5 | 46.3 | 11.4 | 41.4 | 51.1 | 36.5 | 20.0 | 53.6 |
| query2doc$_{Contriever}$ | 47.8 | 32.4 | 7.4 | 27.6 | 29.1 | 17.0 | 9.2 | 29.0 |
| query2doc$_{Contriever-msmarco}$ | 65.2 | 50.6 | 12.0 | 41.8 | **55.9** | **40.4** | **20.1** | 52.2 |
| *Supervised Fine-tuning* | | | | | | | | |
| DPR$^{FT}$ | 40.9 | 26.4 | 5.2 | 16.4 | 22.5 | 13.2 | 5.5 | 15.3 |
| ANCE$^{FT}$ | 56.2 | 40.1 | 8.3 | 25.3 | 42.5 | 26.5 | 11.1 | 31.6 |
| Contriever$^{FT}$ | 53.7 | 38.4 | 8.2 | 23.7 | 35.5 | 20.8 | 8.5 | 22.5 |
| Contriever-msmarco$^{FT}$ | 55.7 | 41.2 | 9.0 | 25.7 | 36.8 | 21.8 | 8.6 | 25.5 |
| *Unsupervised Fine-tuning* | | | | | | | | |
| DPR | 30.1 | 17.5 | 3.8 | 15.6 | 19.6 | 11.2 | 4.8 | 14.3 |
|   w/ INSTRUCTOR$_{QRPG}$ | 50.5 $^{\uparrow20.4}$ | 32.9 $^{\uparrow15.4}$ | 7.2 $^{\uparrow3.4}$ | 24.2 $^{\uparrow8.6}$ | 33.1 $^{\uparrow13.5}$ | 19.4 $^{\uparrow8.2}$ | 9.9 $^{\uparrow5.1}$ | 27.3 $^{\uparrow13.0}$ |
| ANCE | 63.1 | 48.7 | 8.8 | 29.9 | 26.5 | 15.7 | 9.7 | 29.9 |
|   w/ INSTRUCTOR$_{QRPG}$ | 61.2 $^{\downarrow1.9}$ | 46.6 $^{\downarrow2.1}$ | 9.7 $^{\uparrow0.9}$ | 34.4 $^{\uparrow4.5}$ | 43.7 $^{\uparrow17.2}$ | 29.6 $^{\uparrow13.9}$ | 13.6 $^{\uparrow3.9}$ | 40.8 $^{\uparrow10.9}$ |
| Contriever | 25.2 | 12.7 | 3.1 | 13.8 | 18.8 | 9.3 | 8.2 | 25.2 |
|   w/ INSTRUCTOR$_{QRPG}$ | 57.7 $^{\uparrow32.5}$ | 39.8 $^{\uparrow27.1}$ | 9.0 $^{\uparrow5.9}$ | 33.9 $^{\uparrow20.1}$ | 36.9 $^{\uparrow18.1}$ | 22.5 $^{\uparrow13.2}$ | 14.0 $^{\uparrow5.8}$ | 41.5 $^{\uparrow16.3}$ |
| Contriever-msmarco | 63.6 | 46.5 | 10.8 | 41.3 | 27.3 | 15.9 | 12.4 | 39.4 |
|   w/ INSTRUCTOR$_{QRPG}$ | **70.4** $^{\uparrow6.8}$ | **55.1** $^{\uparrow8.6}$ | **13.2** $^{\uparrow2.4}$ | **45.4** $^{\uparrow4.1}$ | 51.7 $^{\uparrow24.4}$ | 32.8 $^{\uparrow16.9}$ | 18.7 $^{\uparrow6.3}$ | **54.0** $^{\uparrow14.6}$ |

Table 2: Experimental results of zero-shot setting on CAsT-19 and CAsT-20. The best results are in **bold**.

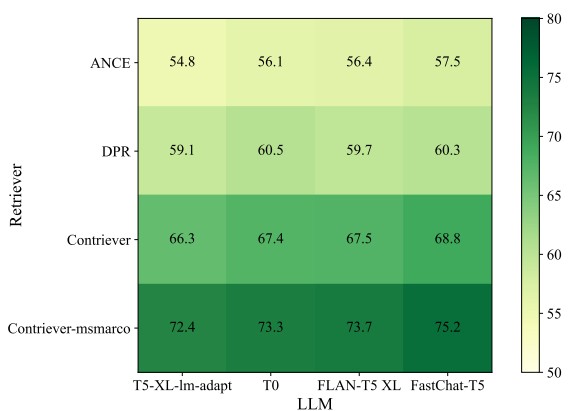

Figure 5: Recall@100 of INSTRUCTOR$_{QRPG}$ on Topi-OCQA with different ad-hoc retrievers and LLMs.

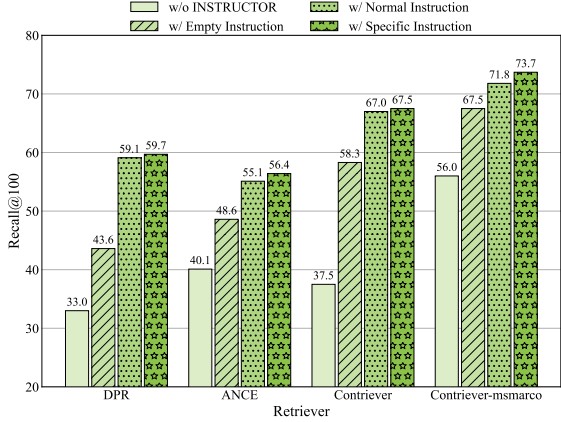

Figure 6: Recall@100 of INSTRUCTOR$_{CRCG}$ on Topi-OCQA with different instructions.

(2) Directly fine-tuning the retrievers on the supervised data shows poor *out-of-distribution* generalization. It suggests that conversational retrieval requires the retrievers to comprehend the conversation rather than relying on retrieval shortcuts.

(3) Unsupervised methods with LLMs like HyDE and query2doc perform well in zero-shot scenarios, indicating the considerable potential of LLMs in zero-shot information retrieval.

### 5.4 Discussion and Analysis

**Selection of Ad-hoc Retrievers and White-box LLMs.** We choose four white-box LLMs (3B): (1) T5-XL-lm-adapt: An improved version of T5;

(2) T0: T5-XL-lm-adapt with instruction tuning; (3) Flan-T5 XL: A more powerful instruction-tuned T5; (4) FastChat-T5: An open-source chatbot fine-tuned from Flan-T5 XL. As illustrated in Figure 5, instruction-tuned LLMs are better instructors for retrievers. In addition, conversational LLMs exhibit enhanced dialogue understanding capabilities and may be more suitable for conversational retrieval.

**Investigation of Different Instructions.** We prompt LLMs with three types of instructions: (1) Empty Instruction: We use meaningless space as the instruction; (2) Normal Instruction: We use "*Please write a text:*" as the instruction; (3) Spe-

| Retriever | All | | First | | No-switch | | Switch | |
|---|---|---|---|---|---|---|---|---|
| | MRR | R@100 | MRR | R@100 | MRR | R@100 | MRR | R@100 |
| Contriever | 6.9 | 37.5 | 8.4 | 60.5 | 8.6 | 42.9 | 2.2 | 17.1 |
| w/ INSTRUCTORCRCG | 16.4 | 67.5 | 17.7 | 80.5 | 19.3 | 71.2 | 8.8 | 54.2 |
| w/ INSTRUCTORQRLV | 18.6 | 71.8 | 19.0 | 81.0 | 21.7 | 75.0 | 11.2 | 61.0 |
| w/ INSTRUCTORQRPG | 27.4 | 78.1 | 27.2 | 85.4 | 27.7 | 79.1 | 20.4 | 71.8 |
| Contriever-msmarco | 16.2 | 56.0 | 36.2 | 89.3 | 17.9 | 60.7 | 6.1 | 34.1 |
| w/ INSTRUCTORCRCG | 23.9 | 73.7 | 38.3 | **89.3** | 27.0 | 75.1 | 15.5 | 60.5 |
| w/ INSTRUCTORQRLV | 28.6 | 80.2 | 34.8 | 84.9 | 31.1 | 84.1 | 19.2 | 65.9 |
| w/ INSTRUCTORQRPG | **38.5** | **83.2** | **46.4** | 87.8 | **39.5** | **84.6** | **33.5** | **78.4** |

Table 3: Experimental results of different question types on TopiOCQA. *All* includes *First*, *No-switch* and *Switch* types. The best results are in **bold**.

cific Instruction: We carefully write instructions for different generation tasks in Appendix D. As depicted in Figure 6, well-designed instructions can help LLMs to calculate relevance scores accurately.

**Scaling with Number of LLM Parameters.** We examine the effect of LLM sizes on retrieval performance improvements. Figure 7 shows the scaling of performance with LLM parameters, including FLAN-T5 small (80M), FLAN-T5 base (250M), FLAN-T5 large (780M) and FLAN-T5 XL (3B). We note that the size of FLAN-T5 is critical for IN-STRUCTORCRCG. So we guess that using a larger FLAN-T5 XXL (11B) or Vicuna (13B) will further improve the retriever's performance. Compared with INSTRUCTORCRCG, INSTRUCTORQRLV has better retrieval performance due to the use of black-box LLM ChatGPT. Overall, the performances of our proposed three instructing strategies continue to improve as the Flan-T5 parameters increase.

**Analysis of Different Question Types.** Following the previous study (Kim and Kim, 2022), we define three question types, *first*, *no-switch*, and *switch*. The *first* question is the first question in conversation without any history. The *no-switch* and *switch* questions can be distinguished by whether $d_t^*$ contains similar or same topics as $d_{t-1}^*$, where the $d_t^*$ is a gold passage at turn $t$ and $t > 1$. We conduct experiments on TopiOCQA to analyze the impact of our approach on different question types. As shown in Table 1, we find that powerful ad-hoc retrievers like Contriever can solve the *first* problem very well, but can hardly handle the *switch* question. After the guidance of our InstructoR, Contriever achieves the most significant performance improvement for the *switch* problem, indicating that LLM-generated supervision signals can help the retriever understand the complex conversation session and discover the user's query intent.

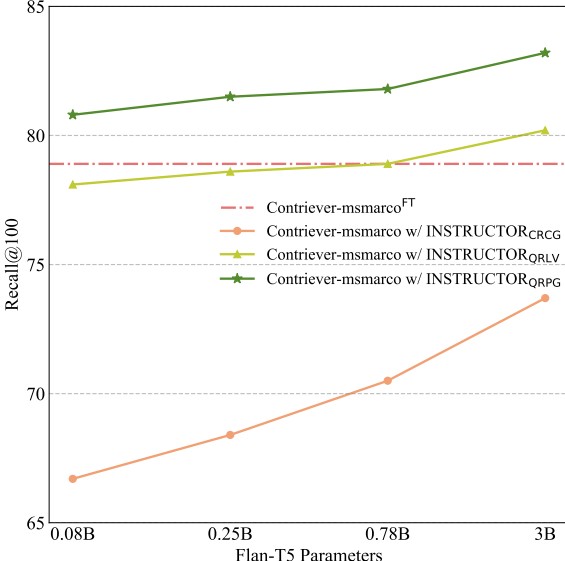

Figure 7: Experimental results with different size of Flan-T5 on TopiOCQA.

Due to the limited space, more comprehensive analysis is shown in Appendix F, G, H, and I.

## 6 Conclusion

In this paper, we propose a novel method termed 🧑‍🏫 INSTRUCTOR to instruct unsupervised conversational dense retrieval with LLMs. INSTRUCTOR calculates the session-passage relevance score with frozen LLMs as the supervised signal to guide the ad-hoc retriever's training. To estimate the relevance score more precisely, we devise three strategies from diverse perspectives: conversational retrieval as conversation generation, question rewrite as latent variable and question response as posterior guide. Experimental results show that INSTRUCTOR can bring significant improvements across various ad-hoc retrievers, even surpassing the current supervised state-of-the-art method.

## Limitations

For further study, we conclude some limitations of our work as follows:

- Limited by computing resources (4 NVIDIA GeForce RTX A6000 GPUs), we can only adopt a white-box LLM (*e.g.*, Flan-T5 XL) with a maximum parameter size of 3B as the instructor. However, we acknowledge that scaling up the LLM's parameters beyond this limit, such as by utilizing a more powerful model like Vicuna (13B) (Chiang et al., 2023) on A100 GPUs, has the potential to further enhance the overall retrieval performance.
- We estimate the relevance score as the conditional generation log likelihood to provide a passage-level supervised signal. In the future, we will explore a more fine-grained relevance estimation method to provide token-level supervision signals.
- Due to the large capability gap between ad-hoc retriever and LLM, this results in the discrepancy of predictions between the retriever and a stronger LLM may tend to be severer. How to mitigate the large discrepancy of predictions remains to be studied.

## Ethics Statement

To ensure the reproducibility of our paper, we will release all source codes, all generated rewrites, and all trained checkpoints upon the acceptance of this paper. We conduct experiments with publicly available conversational retrieval datasets and open-source white-box LLMs. Since our method leverages LLMs to instruct the training of the retrievers, it is important to note that the retrieval results obtained may contain biases from the LLMs. Additionally, we also employ black-box LLMs which can be accessed through OpenAI APIs. However, when considering real-world applications, it is crucial to recognize the potential risks associated with uploading privacy-sensitive conversation data. These factors should be taken into careful consideration for future research and work.

## Acknowledgements

This work is supported by the National Key Research and Development Program of China (No. 2020AAA0106400), the National Natural Science Foundation of China (No. 61976211, 62176257). This work is also supported by the Strategic Priority Research Program of Chinese Academy of Sciences (Grant No.XDA27020100 ), the Youth Innovation Promotion Association CAS, and Yunnan Provincial Major Science and Technology Special Plan Projects (No.202202AD080004).

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

# A  Looking Deeper into Conversational Retrieval

In this paper, we conduct an in-depth investigation of conversational retrieval on QReCC and Topi-OCQA datasets. We employ four ad-hoc retrievers: DPR, ANCE, Contriever and Contriever-msmarco, without fine-tuning them on the conversational retrieval data. To analyze the influence of different input formats on conversational retrieval performance, we conduct experiments with 12 kinds of input formats:

(1) Question: the question $q_t$ at the current turn $t$;

(2) Context: the historical conversation context $c_{t-1}$ before turn $t$;

(3) Context&Question: the sequential concatenation of the context $c_{t-1}$ and question $q_t$;

(4) Question&Context: the reverse concatenation of the context $c_{t-1}$ and question $q_t$;

(5) Answer: the response $a_t$ to question $q_t$;

(6) Answer&Question&Context: the reverse concatenation of the context $c_{t-1}$, question $q_t$ and response $a_t$;

(7) Flan-T5 XL Rewrite: a query rewrite $r_t$ of question $q_t$ generated by Flan-T5 XL[2] (3B) (Chung et al., 2022);

(8) Flan-T5 XXL Rewrite: a query rewrite $r_t$ of question $q_t$ generated by Flan-T5 XXL[3] (11B) (Chung et al., 2022);

(9) T0pp Rewrite: a query rewrite $r_t$ of question $q_t$ generated by T0pp[4] (11B) (Sanh et al., 2022);

(10) Vicuna Rewrite: a query rewrite $r_t$ of question $q_t$ generated by Vicuna[5] (13B);

(11) ChatGPT Rewrite: a query rewrite $r_t$ of question $q_t$ generated by `gpt-3.5-turbo`[6];

(12) Manual Rewrite: a human-rewritten query rewrite $r_t^*$ of question $q_t$, only annotated in QReCC dataset.

---

[2] https://huggingface.co/google/flan-t5-xl/
[3] https://huggingface.co/google/flan-t5-xxl/
[4] https://huggingface.co/bigscience/T0pp/
[5] https://github.com/lm-sys/FastChat/
[6] https://platform.openai.com/docs/models/gpt-3-5

The experimental results on QReCC and Topi-OCQA are presented in Table 4, where we have the following observations:

(1) We provide evidence that *there exists a retrieval shortcut in conversational retrieval*. Following Kim and Kim (2022), we define the shortcut in conversational search as where gold passage $d^*$ can be predicted in top-$K$ predictions even without the current question $q_t$. We find that just using the conversation context gains good performance in QReCC, which suggests that the ad-hoc retrievers cannot really understand the user's query intent, but rely on spurious lexical cues to predict relevant passages. However, the performance of ad-hoc retrievers will drop sharply in TopiOCQA. This is because there are fewer retrieval shortcuts in the scene of topic switching.

(2) We reveal that *conversational query rewriting is not a silver bullet for conversational retrieval*. Query rewrites are not as good as directly using the entire conversation sessions for retrieval in QReCC. Nevertheless, experimental results on TopiOCQA demonstrate query rewriting works well in the scene of topic switching.

(3) We argue that *black-box LLMs have more powerful linguistic capabilities than white-box LLMs*. We provide several query rewriting examples in Table 5, 6, 7, 8, 9 and 10. Black-box LLM ChatGPT achieves human-comparable or even better query rewriting quality in QReCC. Sometimes ChatGPT is too smart and will answer questions redundantly (*e.g.*, Figure 8) or refuse to rewrite uncertain questions (*e.g.*, Figure 10). For white-box LLMs, the effect of query rewriting continues to improve as the LLM parameters increase. However, T0pp may not have been sufficiently fine-tuned with instruction data, so it is difficult to follow our instructions to generate rewrites. It is worth mentioning that Vicuna which is an open-source chatbot trained by fine-tuning LLaMA (Touvron et al., 2023) achieves the closest effect to ChatGPT.

(4) We find that *question responses can provide the relevant signal when the gold passages are unknown*. Compared with Question&Context, using Answer&Question&Context as the input format has a significant improvement on QReCC and TopiOCQA. Inspire by this, we treat question responses as the posterior guide to further enhance the training of ad-hoc retrievers.

(5) We find a trick that *the reverse concatenation of conversation sessions leads to better retrieval*

*performance*. Context&Question shows better results than Question&Context on all four retrievers. Due to the limited context of the ad-hoc retrievers, We use reverse concatenation to avoid losing the important information in the latest conversation.

## B  Related Work

### B.1  Conversational Retrieval

Different from traditional single-turn ad-hoc retrieval, conversational retrieval needs to deal with the multi-turn conversation context and understand the user's query intent. To solve conversational retrieval, existing methods are mainly divided into two categories: conversational query rewriting (Yu et al., 2020; Lin et al., 2020; Vakulenko et al., 2021; Qian and Dou, 2022; Mo et al., 2023a,b; Mao et al., 2023a) and conversational dense retrieval (Yu et al., 2021; Lin et al., 2021; Mao et al., 2022a; Ishii et al., 2022; Mao et al., 2022b, 2023b; Hai Le et al., 2023).

Conversational query rewriting methods utilize the rewriting models to explicitly reformulate the conversational queries into de-contextualized rewrites and then perform standard ad-hoc retrieval. Yu et al. (2020) first large amounts of ad hoc search sessions to generate weak supervision data, then fine-tune GPT-2 to rewrite conversational queries. Lin et al. (2020) demonstrate the effectiveness of T5 to rewrite queries. For better QA performance, Chen et al. (2022) propose using QA feedback to supervise the rewriting model with reinforcement learning. Wu et al. (2022) also adopt reinforcement learning to directly optimize the rewritten query towards retrieval performance. Recently, Qian and Dou (2022) propose a unified framework for query rewriting and context modelling. Mao et al. (2023a) presents a prompting framework called LLMCS that leverages LLMs to perform few-shot conversational query rewriting for conversational retrieval.

Conversational dense retrieval methods usually fine-tune the ad-hoc retrievers on a large number of labeled session-passage pairs. Due to the lack of supervised data, Yu et al. (2021) fine-tune the ad-hoc search retriever on conversational search data using knowledge distillation for few-shot retrieval. Kim and Kim (2022) train the dense retriever with hard negatives effectively mitigates the heavy shortcut dependency. Lin et al. (2021) leverage external datasets to produce more pseudo-relevance labels for conversational search to overcome the lack of training data. Mao et al. (2022a)

use contrastive learning to train the conversational query encoder for context denoising. Recently, Mao et al. (2023b) propose a sparse lexical-based conversational retriever two well-matched multi-level denoising methods.

### B.2  LLM for Information Retrieval

Recently, some research (Sachan et al., 2022a,b; Yu et al., 2023; Saad-Falcon et al., 2023; Jagerman et al., 2023; Sun et al., 2023; Shen et al., 2023; Shi et al., 2023; Mao et al., 2023a; Qin et al., 2023) has focused on applying LLMs (Brown et al., 2020; Sun et al., 2022; Gu et al., 2023; Workshop et al., 2022; Ouyang et al., 2022; OpenAI, 2023) in information retrieval tasks. Most of these works use LLMs to generate query-related documents to expand the original query in few-shot or zero-shot ad-hoc retrieval scenarios. Dai et al. (2022) prompt LLMs to generate synthetic task-specific training data for few-shot retrieval. Sachan et al. (2022a) use LLMs as the zero-shot reranker to improve passage retrieval in open domain question answering. Given a query, Gao et al. (2022) first use LLMs to generate the hypothetical document, then ground the generated document to the actual corpus. Wang et al. (2023) first generate pseudo documents by few-shot prompting LLMs, and then expands the query with generated pseudo documents. Sun et al. (2023) evaluate the capabilities of ChatGPT and GPT-4 on various passage reranking benchmarks.

In this paper, we consider a more realistic setting which is to train a conversational dense retriever in an *unsupervised* manner, *i.e.*, without using session-passage pairs. We only have a certain number of conversation sessions and a large passage collection during the training. To the best of our knowledge, this is the first attempt to leverage LLMs to guide the training of unsupervised conversational retrieval.

## C  Dataset Details

Table 11 shows the statistics of QReCC, TopiOCQA, CAsT-19, and CAsT-20 datasets.

**QReCC** is the first large-scale open-domain conversational QA dataset that contains human-annotated question rewrites. QReCC contains conversations from QuAC (Choi et al., 2018), TREC CAsT and Natural Questions (Kwiatkowski et al., 2019). In this paper, we focus on solving the conversational retrieval task.

**TopiOCQA** is an open-domain conversational QA dataset with topic switches based on Wikipedia. On average, a conversation in TopiOCQA has 13 question-answer turns and involves 4 topics. In contrast to QReCC, TopiOCQA does not provide human-annotated query rewrites.

**CAsT-19 and CAsT-20** are two conversational search datasets released by TREC Conversational Assistance Track (CAsT). Due to the limited number of conversations contained in them, they are often used as the evaluation datasets.

## D Instruction Templates

### D.1 Conversational Retrieval as Conversation Generation

---

$I_z^c$: Context Generation

---

Passage: {passage}.
Please create a conversation (consisting of several questions and answers) based on the given passage:
Conversation context: {context}.

---

$I_{z,c}^q$: Conversational Query Generation

---

Passage: {passage}.
Conversation context: {context}.

Please complete the conversation with a question based on the given passage and conversation context:
Question: {question}.

---

### D.2 Question Rewrite as Latent Variable

---

$I_{c,q}^r$: Query Rewrite Generation

---

You are a helpful assistant that need to understand the following conversation.
Conversation context: {context}.
Question: {question}.
To better retrieve the relevant passages, please reformulate the last question of the given conversation context into a complete and de-contextualized question rewrite:

---

$I_z^r$: Question Generation

---

Passage: {passage}.

Please write a detailed question based on the given passage:

Question: {question}.

---

### D.3 Question Response as Posterior Guide

---

$I_{z,c,q}^a$: Response Generation

---

Passage: {passage}.
Conversation context: {context}.
Please answer the following question based on the given passage and conversation context:
Question: {question}.
Answer: {answer}.

---

## E Parameter Settings

Our implementation is based on HuggingFace's Transformers[7], Sentence Transformers[8], Megatron[9] and PyTorch[10]. For ad-hoc retrievers, we adopt DPR[11], ANCE[12], Contriever[13], and Contriever-msmarco[14]. In our unsupervised training framework, we retrieve the top-$K = 32$ passages to train the ad-hoc retriever. We set the value of the temperature hyperparameter $\tau$ using cross-validation. We use the Adam algorithm to optimize retriever parameters. The learning rate is initialized as 1e-6 with warmup and linear decay. The batch size is set to 4. For INSTRUCTOR$_{\text{QRLV}}$, we use the manual query rewrites in QReCC and generated query rewrites in TopiOCQA. We sample 15,000 conversations in the TopiOCQA training set and use gpt-3.5-turbo to generate $|\mathcal{R}| = 3$ possible rewrites for each conversation with a temperature is 0.7. To ensure rewriting quality, we use regular expressions to filter out those rewrites

---

[7]https://github.com/huggingface/transformers/
[8]https://www.sbert.net/
[9]https://github.com/NVIDIA/Megatron-LM/
[10]https://github.com/pytorch/pytorch/
[11]https://huggingface.co/sentence-transformers/facebook-dpr-ctx_encoder-single-nq-base/
[12]https://huggingface.co/sentence-transformers/msmarco-roberta-base-ance-firstp/
[13]https://huggingface.co/facebook/contriever/
[14]https://huggingface.co/facebook/contriever-msmarco/

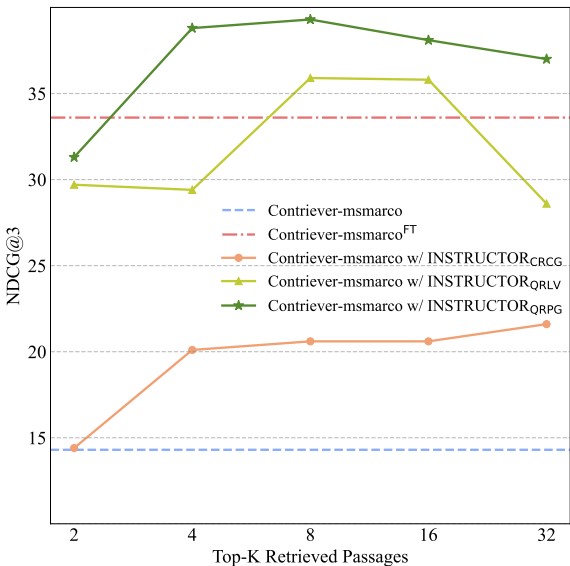

Figure 8: NDCG@3 of different strategies with different numbers $K$ of retrieved passages on TopiOCQA.

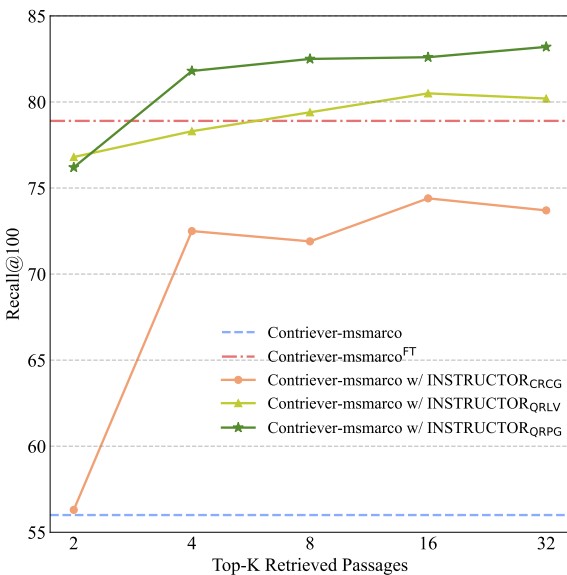

Figure 9: Recall@100 of different strategies with different numbers $K$ of retrieved passages on TopiOCQA.

that follow our instructions incorrectly. We adopt the gold responses in our main experiments for INSTRUCTOR$_{\text{QRPG}}$. We also use `gpt-3.5-turbo` to generate query responses with a temperature is 0.3. All experiments are conducted with 4 NVIDIA GeForce RTX A6000 GPUs.

## F   Effect of Top-$K$ Retrieved Passages

We conduct experiments with different numbers ($K = 2, 4, 8, 16, 32$) of retrieved passages during training. As shown in Figure 8 and 9, we find that a smaller number of retrieved passages leads to better NDCG@3 performance, and a larger number of retrieved passages produces better Recall@100 performance. But retrieving too small passages, like $K = 2$, doesn't yield reasonable results. In this paper, we set $K = 32$ which offers a reasonable middle ground, while $K = 16$ may lead to better performance.

## G   Ablation of Question Rewrites

We conduct an ablation study on INSTRUCTOR$_{\text{QRLV}}$ with Vicuna Rewrite and ChatGPT Rewrite. Vicuna Rewrite denotes the query rewrites generated by Vicuna, and ChatGPT Rewrite denotes the query rewrites generated by `gpt-3.5-turbo`. As shown in Figure 10, ChatGPT Rewrite performs better than Vicuna Rewrite because of the higher rewriting quality of ChatGPT. Besides, we find that the number $|\mathcal{R}|$ of possible rewrites is not the higher the better. This may be

because larger $\mathcal{R}$ will contain incorrect rewrites, introducing more noise. Hence, we set $|\mathcal{R}|$ to a moderate value of 3 in our main experiments.

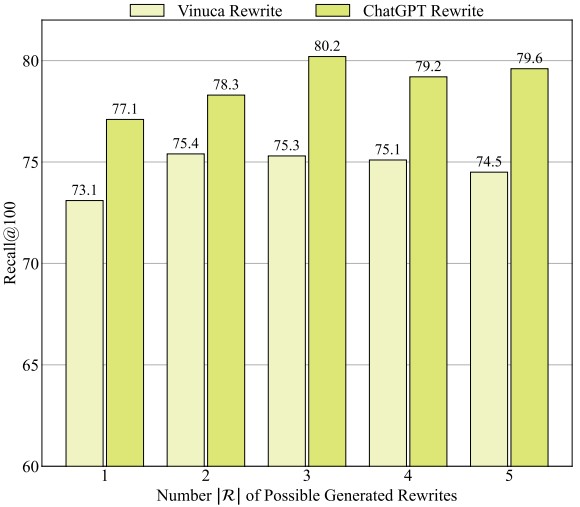

Figure 10: Recall@100 of Contriever-msmarco w/ INSTRUCTOR$_{\text{QRLV}}$ with Vicuna Rewrite and ChatGPT Rewrite on TopiOCQA.

## H   Ablation of Question Responses

We conduct an ablation study on INSTRUCTOR$_{\text{QRPG}}$ with Gold Response and Generated Response. Gold Response denotes the real response in the original conversation, and Generated Response denotes the response generated by `gpt-3.5-turbo`. As illustrated in Figure 11, Generated Response works better than Gold Response when there are a

small number of training samples. However, when there are enough training samples, Gold Response has better performance than Generated Response, especially in NDCG@3. Hence, We adopt the Gold Response for INSTRUCTOR_QRPG in our main experiments.

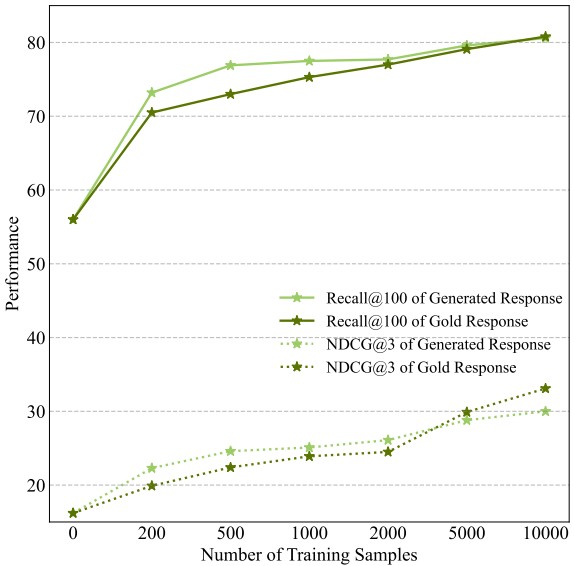

Figure 11: Recall@100 and NDCG@3 of Contriever-msmarco w/ INSTRUCTOR_QRPG with Manual Response and Generated Response on TopiOCQA.

## I Recall@$N$ of Different Strategies

We also present the Recall@$N$ of different instructing strategies with different numbers $N = \{1, 3, 5, 10, 20, 50, 100\}$ of retrieved passages in Figure 12. We can find that Contriever-msmarco and INSTRUCTOR_QRPG are the best combination.

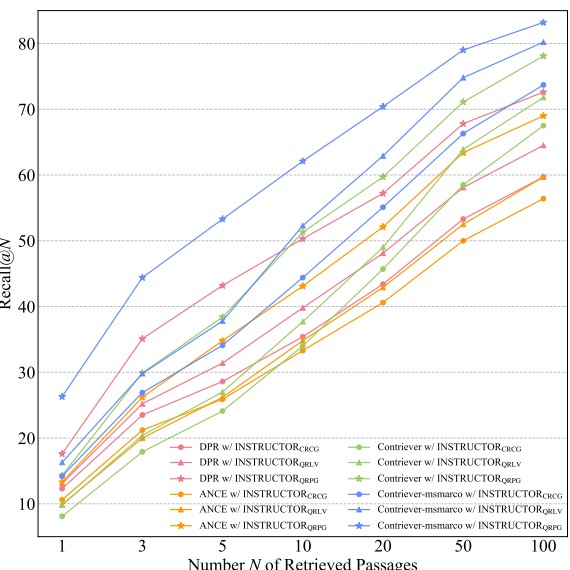

Figure 12: Recall@$N$ of different strategies with different numbers $N$ of retrieved passages on TopiOCQA.

| Retriever | Input Format | QReCC | | | | TopiOCQA | | | |
|---|---|---|---|---|---|---|---|---|---|
| | | MRR | NDCG@3 | R@10 | R@100 | MRR | NDCG@3 | R@10 | R@100 |
| DPR | Question | 5.5 | 4.8 | 8.4 | 14.3 | 3.7 | 3.3 | 6.7 | 12.2 |
| | Context | 24.9 | 23.1 | 37.2 | 51.4 | 7.4 | 6.9 | 12.6 | 23.5 |
| | Context&Question | 29.7 | 27.4 | 44.1 | 61.5 | 11.5 | 10.6 | 19.7 | 35.0 |
| | Question&Context | 28.7 | 26.3 | 43.0 | 60.7 | 10.7 | 9.8 | 18.2 | 33.0 |
| | Answer | 24.5 | 22.9 | 34.8 | 51.1 | 15.5 | 15.1 | 21.5 | 30.0 |
| | Answer&Question&Context | 36.7 | 34.3 | 53.3 | 70.5 | 14.5 | 13.4 | 24.2 | 40.2 |
| | Flan-T5 XL Rewrite | 12.1 | 10.7 | 19.1 | 31.5 | 11.7 | 10.7 | 21.5 | 35.0 |
| | Flan-T5 XXL Rewrite | 13.4 | 11.8 | 20.9 | 33.9 | 16.1 | 15.4 | 27.7 | 41.7 |
| | T0pp Rewrite | 12.6 | 11.1 | 20.1 | 32.4 | 8.6 | 7.8 | 15.2 | 25.8 |
| | Vicuna Rewrite | 14.1 | 12.4 | 22.2 | 35.9 | 14.3 | 13.2 | 25.8 | 44.1 |
| | ChatGPT Rewrite | 21.6 | 19.3 | 33.7 | 51.5 | 15.4 | 14.3 | 27.6 | 46.2 |
| | Manual Rewrite | 21.7 | 19.5 | 33.1 | 51.0 | - | - | - | - |
| ANCE | Question | 9.8 | 9.0 | 15.0 | 20.1 | 3.7 | 3.4 | 7.2 | 13.1 |
| | Context | 30.9 | 29.0 | 47.1 | 58.8 | 6.1 | 5.4 | 10.7 | 21.8 |
| | Context&Question | 38.2 | 35.7 | 57.6 | 72.3 | 8.7 | 7.8 | 15.6 | 29.9 |
| | Question&Context | 41.4 | 38.9 | 61.2 | 74.7 | 11.6 | 10.2 | 21.8 | 40.1 |
| | Answer | 40.5 | 38.6 | 54.0 | 63.9 | 10.5 | 9.9 | 16.1 | 26.6 |
| | Answer&Question&Context | 54.0 | 51.6 | 72.9 | 83.5 | 23.2 | 21.7 | 37.6 | 55.8 |
| | Flan-T5 XL Rewrite | 22.3 | 20.4 | 34.4 | 46.1 | 14.0 | 13.3 | 25.0 | 36.4 |
| | Flan-T5 XXL Rewrite | 20.5 | 18.6 | 32.1 | 43.4 | 15.9 | 14.7 | 27.8 | 40.8 |
| | T0pp Rewrite | 19.9 | 18.2 | 30.9 | 41.4 | 9.0 | 8.4 | 16.1 | 25.7 |
| | Vicuna Rewrite | 28.7 | 26.6 | 43.8 | 57.3 | 19.6 | 18.4 | 34.6 | 51.9 |
| | ChatGPT Rewrite | 32.5 | 29.9 | 49.3 | 63.5 | 22.7 | 21.7 | 39.3 | 58.0 |
| | Manual Rewrite | 33.7 | 31.2 | 50.2 | 65.3 | - | - | - | - |
| Contriever | Question | 6.1 | 5.0 | 11.2 | 20.3 | 1.5 | 1.1 | 3.4 | 9.7 |
| | Context | 37.1 | 35.2 | 55.0 | 69.9 | 5.8 | 4.8 | 11.7 | 28.2 |
| | Context&Question | 41.2 | 38.4 | 63.2 | 83.4 | 5.8 | 4.1 | 12.3 | 34.1 |
| | Question&Context | 41.5 | 38.7 | 63.8 | 84.3 | 6.9 | 5.3 | 14.5 | 37.5 |
| | Answer | 56.5 | 54.8 | 71.7 | 83.4 | 21.4 | 20.5 | 29.8 | 42.2 |
| | Answer&Question&Context | 58.9 | 57.0 | 80.3 | 92.4 | 19.0 | 16.9 | 34.7 | 60.7 |
| | Flan-T5 XL Rewrite | 16.1 | 13.7 | 28.4 | 49.6 | 5.8 | 4.6 | 12.5 | 29.2 |
| | Flan-T5 XXL Rewrite | 15.5 | 13.1 | 27.5 | 49.2 | 7.3 | 5.8 | 15.0 | 35.3 |
| | T0pp Rewrite | 21.1 | 18.9 | 35.3 | 53.7 | 4.9 | 3.8 | 11.1 | 25.7 |
| | Vicuna Rewrite | 23.3 | 20.5 | 40.4 | 65.5 | 9.8 | 8.0 | 21.4 | 45.6 |
| | ChatGPT Rewrite | 28.7 | 25.5 | 48.6 | 74.9 | 12.4 | 10.3 | 26.2 | 52.4 |
| | Manual Rewrite | 26.9 | 23.5 | 46.3 | 74.6 | - | - | - | - |
| Contriever -msmarco | Question | 11.5 | 10.3 | 18.4 | 27.1 | 5.1 | 4.7 | 9.2 | 16.2 |
| | Context | 36.6 | 34.6 | 55.0 | 70.4 | 7.8 | 6.8 | 14.0 | 29.8 |
| | Context&Question | 44.9 | 42.2 | 68.1 | 86.6 | 11.6 | 10.1 | 20.6 | 40.8 |
| | Question&Context | 47.6 | 45.0 | 70.5 | 88.9 | 16.2 | 14.3 | 29.4 | 56.0 |
| | Answer | 56.3 | 54.7 | 71.4 | 82.5 | 25.5 | 24.9 | 34.5 | 47.1 |
| | Answer&Question&Context | 63.0 | 61.2 | 82.4 | 93.8 | 32.1 | 30.6 | 52.4 | 75.1 |
| | Flan-T5 XL Rewrite | 29.6 | 26.8 | 47.0 | 66.1 | 17.8 | 16.9 | 30.6 | 43.6 |
| | Flan-T5 XXL Rewrite | 27.1 | 24.4 | 43.9 | 62.7 | 20.0 | 19.1 | 34.2 | 48.2 |
| | T0pp Rewrite | 26.5 | 24.2 | 42.4 | 59.3 | 11.8 | 10.9 | 19.9 | 32.4 |
| | Vicuna Rewrite | 38.0 | 35.0 | 59.3 | 79.4 | 26.0 | 24.5 | 45.0 | 63.9 |
| | ChatGPT Rewrite | 44.0 | 40.8 | 67.0 | 87.1 | 30.0 | 28.3 | 51.3 | 72.3 |
| | Manual Rewrite | 44.1 | 40.7 | 67.6 | 88.6 | - | - | - | - |

Table 4: Experimental results of different input formats and ad-hoc retrievers on QReCC and TopiOCQA.

| | |
|---|---|
| Conversation Session | $q_1$: What happened to Martha and the Vandellas during motown major hit years?
$a_1$: Following their signing to Motown's Gordy imprint in 1962, the Vandellas struck gold with their second release.
$q_2$: What happened after that?
$a_2$: Martha and the Vandellas' second hit, Heat Wave, became a phenomenal record for the group.
$q_3$: Can you tell me how long did it stay on the charts?
$a_3$: Martha and the Vandellas' Heat Wave, became a phenomenal record, hitting number one on the R&B singles chart for five weeks.
$q_4$: What were some of their major hits?
$a_4$: Martha and the Vandellas' string of hits included Come and Get These Memories, Heat Wave, and Dancing in the Street.
$q_5$: Were they popular among a lot of people?
$a_5$: Martha and the Vandellas struck gold with their second release, titled Come and Get These Memories.
$q_6$: What other things took place after that?
$a_6$: Heat Wave became their first million-seller and eventually got the group their only Grammy Award nomination.
$q_7$: Did they go on to continue to be successful?
$a_7$: Martha and the Vandellas' success continued with their second Top Ten single and third Top 40 single, Quicksand.
$q_8$: Any other important aspects regarding their major hit years?
$a_8$: Annette, who was pregnant with her first child and set to get married, chose to leave her singing career behind by 1964.
$q_9$: Did the group grow apart after she left?
$a_9$: Betty Kelley, formerly of the Velvelettes, was brought in shortly afterward.
$q_{10}$: **Did she do well with joining the group?** |
| Flan-T5 XL Rewrite | Response: Betty Kelley, formerly of the Velvelettes, was brought in shortly |
| Flan-T5 XXL Rewrite | Can you tell me how long it stayed on the charts? |
| T0pp Rewrite | Martha and the Vandellas' second hit, Heat Wave, became a phenomenal record. |
| Vicuna Rewrite | Question: What happened to Martha and the Vandellas after Annette left the group and Betty Kelley joined? |
| ChatGPT Rewrite | How did Betty Kelley perform after joining Martha and the Vandellas? |
| Manual Rewrite | Did Betty Kelley do well with joining the group Martha and the Vandellas? |
| Gold Response | Kelley was let go after reportedly missing shows, as well as getting into altercations with Reeves. |

Table 5: QReCC example 1. Red denotes the current user's query. Blue denotes the correct rewritten content.

| | |
|---|---|
| Conversation Session | $q_1$: When was the expansion for the Arapaho?
$a_1$: Once established, the Arapaho began to expand on the plains through trade, warfare, and alliances with other plains tribes. Around 1811, the Arapaho made an alliance with the Cheyenne.
$q_2$: What expanded?
$a_2$: Their strong alliance with the Cheyenne allowed the Arapaho to greatly expand their hunting territory.
$q_3$: What was special about the expansion?
$a_3$: Conflict with the allied Comanche and Kiowa ended in 1840 when the two large tribes made peace with the Arapaho and Southern Cheyenne and became their allies.
$q_4$: Who did they trade with?
$a_4$: Along the upper Missouri River, the Arapaho actively traded with the farming villages of the Arikara, Mandan, and Hidatsa.
$q_5$: Who did they go to war with?
$a_5$: By 1826, the Lakota, Dakota, Cheyenne, and Arapaho pushed the Kiowa (Niiciiheihiinenno\u02bc; Kiowa tribe: Niiciiheihiiteen) and invading Comanche to the south.
$q_6$: **What types of things did they trade?** |
| Flan-T5 XL Rewrite | Response: What types of things did the Arapaho trade? |
| Flan-T5 XXL Rewrite | Can you tell me how long it stayed on the charts? |
| T0pp Rewrite | What did the Arapaho trade with the Arikara, Mandan, and Hidatsa. |
| Vicuna Rewrite | Question: What did the Arapaho trade with the farming villages along the upper Missouri River?\nResponse: Along the upper Missouri River, the Arapaho actively traded with the farming villages of the Arikara, Mandan, and Hidatsa. |
| ChatGPT Rewrite | What were the items or goods that the Arapaho actively traded with the farming villages of the Arikara, Mandan, and Hidatsa along the upper Missouri River? |
| Manual Rewrite | What types of things did the Arapaho trade? |
| Gold Response | They traded meat and hides for corn, squash, and beans. |

Table 6: QReCC example 2. **Red** denotes the current user's query. Blue denotes the correct rewritten content.

| | |
|---|---|
| Conversation Session | $q_1$: Tell me about William Dampier's second circumnavigation?
$a_1$: William Dampier was appointed commander of the 26-gun ship St George, with a crew of 120 men.
$q_2$: Where did he set sail from?
$a_2$: William Dampier set sail on 11 September 1703 from Kinsale, Ireland.
$q_3$: Did he encounter pirates?
$a_3$: UNANSWERABLE.
$q_4$: Where did they go from Kinsale?
$a_4$: The two ships made a storm-tossed passage round Cape Horn, arriving at the Juan Fern\u00e1ndez Islands off the coast of Chile in February 1704.
$q_5$: What did they do there?
$a_5$: While watering and provisioning there, they sighted a heavily armed French merchantman, which they engaged in a seven-hour battle but were driven off.
$q_6$: **Did they fight?** |
| Flan-T5 XL Rewrite | Response: Did William Dampier fight with the French? |
| Flan-T5 XXL Rewrite | What was the name of the French merchantman? |
| T0pp Rewrite | Did they fight? |
| Vicuna Rewrite | Question rewrite:\nWhat happened during the seven-hour battle between the St George and a heavily armed French merchantman that they engaged off the coast of Chile? |
| ChatGPT Rewrite | Did William Dampier and his crew engage in a battle with a French merchantman while at the Juan Fern\u00e1ndez Islands? |
| Manual Rewrite | Did William Dempier and his crew fight in Chile? |
| Gold Response | Yes, they engaged in a seven-hour battle but were driven off. |

Table 7: QReCC example 3. Red denotes the current user's query. Blue denotes the correct rewritten content.

| | |
|---|---|
| Conversation Session | $q_1$: Does callie baby die in season 7 episode 18? 
 $a_1$: No. 
 $q_2$: Who plays the character mentioned above? 
 $a_2$: Sara Ramirez. 
 $q_3$: Apart from acting, does she have a career in any other profession? 
 $a_3$: She is a singer and songwriter. 
 $q_4$: Name some of her songs? 
 $a_4$: Silent Night. 
 $q_5$: What is the significance of the above song? 
 $a_5$: It is a popular Christmas carol. 
 $q_6$: Who has written it? 
 $a_6$: Joseph Mohr. 
 $q_7$: The above mentioned episode is from which series? 
 $a_7$: Grey's Anatomy. 
 $q_8$: **Name some characters of it.** |
| Flan-T5 XL Rewrite | Response: what are some characters in the show grey's anatomy? |
| Flan-T5 XXL Rewrite | What is the name of the character who dies in season 7 episode 18? |
| T0pp Rewrite | Turn 3: Sara Ramirez is a singer and songwriter. What is the significance |
| Vicuna Rewrite | Question: Can you give me the names of some characters of the TV series Grey's Anatomy? |
| ChatGPT Rewrite | Rewrite: Can you provide some names of the characters from the TV series Grey's Anatomy? \nResponse: Meredith Grey, Derek Shepherd, Miranda Bailey, Alex Karev, Cristina Yang, and many more. |
| Gold Passage | Grey's Anatomy: Introduction. Justin Chambers, who played Dr. Alex Karev, left the show during this season. \"Grey's Anatomy\" is the longest-running scripted primetime show currently airing on ABC, and the longest scripted primetime series carried by ABC. As of February 28, 2019, it was the longest running American primetime medical drama series. The series was most recently renewed for its seventeenth season. Its success catapulted longtime series regulars including Pompeo, Dempsey and Oh to worldwide recognition; they were among the top 5 highest-earning television actors in 2013. While the show's ratings have fallen over the course of its run (it was once among the overall top 10 shows in the United States), it is still one of the highest-rated shows among the 18\u201349 demographic, and the #3 drama on all of broadcast television in the United States. |
| Gold Response | Meredith Grey, Alex Karev, Miranda Bailey and Richard Webber. |

Table 8: TopiOCQA example 1. **Red** denotes the current user's query. Blue denotes the correct rewritten content.

| | |
|---|---|
| Conversation Session | $q_1$: Which was the new building constructed at the start of the 1900s? |
| | $a_1$: It transformed its society into the world's first industrialised nation. |
| | $q_2$: Which age can its history be traced back to? |
| | $a_2$: 780,000 years ago. |
| | $q_3$: What are some of the rivers found in this country? |
| | $a_3$: Thames, Mersey, Tyne, Severn etc. |
| | $q_4$: Any well-known museums here? |
| | $a_4$: London's British Museum. |
| | $q_5$: Where is this museum located? |
| | $a_5$: Bloomsbury area of London, United Kingdom. |
| | $q_6$: Who was the founder? |
| | $a_6$: Sir Hans Sloane. |
| | $q_7$: **Which was the new building constructed at the start of the 1900s?** |
| Flan-T5 XL Rewrite | Response: what was the new building constructed at the start of the 1900s? |
| Flan-T5 XXL Rewrite | What was the new building constructed at the start of the 1900s? |
| T0pp Rewrite | The British Museum is located in the Bloomsbury area of London, United Kingdom. |
| Vicuna Rewrite | Question rewrite:\nWhat was the name of the building that was constructed at the start of the 1900s at the British Museum? |
| ChatGPT Rewrite | Can you provide information about the new building constructed at the start of the 1900s in the British Museum located in the Bloomsbury area of London, United Kingdom? |
| Gold Passage | British Museum: New century, new building (1900\u20131925). Morgan had also acquired a major part of Sir John Evans's coin collection, which was later sold to the museum by his son John Pierpont Morgan Junior in 1915. In 1918, because of the threat of wartime bombing, some objects were evacuated via the London Post Office Railway to Holborn, the National Library of Wales (Aberystwyth) and a country house near Malvern. On the return of antiquities from wartime storage in 1919 some objects were found to have deteriorated. A conservation laboratory was set up in May 1920 and became a permanent department in 1931. It is today the oldest in continuous existence. |
| Gold Response | A conservation laboratory. |

Table 9: TopiOCQA example 2. Red denotes the current user's query. Blue denotes the correct rewritten content.

| | |
|---|---|
| Conversation Session | $q_1$: Which of the following is the ultimate result of evolution?
$a_1$: UNANSWERABLE.
$q_2$: What is evolution?
$a_2$: It is change in the heritable characteristics of biological populations over successive generations.
$q_3$: Who was the first to study this?
$a_3$: Charles Darwin and Alfred Russel Wallace.
$q_4$: When was the first one born?
$a_4$: Shrewsbury, Shropshire.
$q_5$: Did he suffer from any illness?
$a_5$: He had "an uncomfortable palpitation of the heart".
$q_6$: Did he publish anything?
$a_6$: Yes, his theory of evolution with compelling evidence in his 1859 book "On the Origin of Species".
$q_7$: What does modern synthesis state about evolution?
$a_7$: It explained patterns observed across species in populations, through fossil transitions in palaeontology, and complex cellular mechanisms in developmental biology.
$q_8$: What is the last type of biology about?
$a_8$: It is the study of the process by which animals and plants grow and develop.
$q_9$: Does it have any model for organisms?
$a_9$: UNANSWERABLE.
$q_{10}$: **Where has the study of evolution been applied to?** |
| Flan-T5 XL Rewrite | Response: What has the study of evolution been applied to? |
| Flan-T5 XXL Rewrite | What is the ultimate result of evolution? |
| T0pp Rewrite | Change in the heritable characteristics of biological populations over successive generations. |
| Vicuna Rewrite | Question rewrite:\nWhat are some areas where the study of evolution has been applied? |
| ChatGPT Rewrite | Please provide a complete and de-contextualized question rewrite for the last question in the conversation. |
| Gold Passage | Evolution: Applications. Concepts and models used in evolutionary biology, such as natural selection, have many applications. Artificial selection is the intentional selection of traits in a population of organisms. This has been used for thousands of years in the domestication of plants and animals. More recently, such selection has become a vital part of genetic engineering, with selectable markers such as antibiotic resistance genes being used to manipulate DNA. Proteins with valuable properties have evolved by repeated rounds of mutation and selection (for example modified enzymes and new antibodies) in a process called directed evolution. Understanding the changes that have occurred during an organism's evolution can reveal the genes needed to construct parts of the body, genes which may be involved in human genetic disorders. |
| Gold Response | Domestication of plants and animals, genetic engineering, etc. |

Table 10: TopiOCQA example 3. **Red** denotes the current user's query. Blue denotes the correct rewritten content.

| Dataset | | # Conversations | # Turns | # Passages |
|---|---|---|---|---|
| QReCC | Train | 10,823 | 63,501 | 54M (54,573,064) |
| | Test | 2,775 | 16,451 | |
| TopiOCQA | Train | 3,509 | 45,450 | 25M (25,700,592) |
| | Test | 205 | 2,514 | |
| CAsT-19 | Test | 50 | 479 | 38M (38,429,852) |
| CAsT-20 | Test | 25 | 216 | |

Table 11: Statistics of conversational retrieval datasets.