# OpenReview forum: "InstructoR: Instructing Unsupervised Conversational Dense Retrieval with Large Language Models"
_EMNLP/2023/Conference — EMNLP 2023 Findings_

### Official Review · Reviewer_3sLe · 2023-08-02

**Soundness:** 4

**Excitement:**

3: Ambivalent: It has merits (e.g., it reports state-of-the-art results, the idea is nice), but there are key weaknesses (e.g., it describes incremental work), and it can significantly benefit from another round of revision. However, I won't object to accepting it if my co-reviewers champion it.

**Paper Topic And Main Contributions:**

This paper proposes a novel method called INSTRUCTOR for unsupervised conversational dense retrieval. The key idea is to leverage large language models (LLMs) to provide supervised signals to guide the training of retrievers, without requiring any labeled data. The main contributions are:

1. An unsupervised training framework where LLMs generate soft session-passage relevance scores to instruct the retrievers.

2. Three strategies to accurately estimate relevance from different perspectives: conversational retrieval as conversation generation, question rewrite as latent variable, and question response as posterior guide.

The paper addresses the lack of labeled data for conversational retrieval, and demonstrates how large pretrained models can be leveraged for unsupervised learning in this setting.

**Questions For The Authors:**

Can you provide more analysis into how the quality of LLM-generated supervision signals varies across different types of conversations or questions? Any patterns in where it works better/worse?

**Reasons To Accept:**

1. The paper tackles an important problem of unsupervised conversational retrieval training. The way using LLMs to provide training signals is a novel in the conversational context (although it has been validated in ad-hoc search), and the three strategies provide nice ways to leverage LLMs from different perspectives of the conversation. This can enable building conversational retrievers without labeled data.

2. Thorough ablation studies analyzing the effect of different proposed strategies.

3. Strong empirical results, surpassing supervised approaches on two datasets.

4. Well-written paper with sufficient details to reproduce the approach.

**Reasons To Reject:**


1. Limited analysis on how different types of conversations/questions affect the quality of LLM supervision.

2. Unclear how the approach scales as the dataset size increases. Memory and compute costs?


**Reproducibility:**

3: Could reproduce the results with some difficulty. The settings of parameters are underspecified or subjectively determined; the training/evaluation data are not widely available.

**Reviewer Confidence:**

3: Pretty sure, but there's a chance I missed something. Although I have a good feel for this area in general, I did not carefully check the paper's details, e.g., the math, experimental design, or novelty.

**Typos Grammar Style And Presentation Improvements:**

Line 143: "discover" -> "discovering"

Line 289: Extra space before period

Section 3.2: Consider using sub-section headings for each strategy

Figure 1: Increase font size of text in the boxes

Figure 3: Improve spacing around "+" symbols

---

> ### Author Rebuttal · Authors · 2023-08-29
>
> Thanks for your careful and insightful reviews. Your professional reviews offer us great advice towards writing a more comprehensive and competitive paper!
>
> > **Response to Rejection 1**: Limited analysis on how different types of conversations/questions affect the quality of LLM supervision.
>
> Thanks for your valuable advice! **We conduct analysis on how different types of conversations/questions affect the quality of LLM supervision**. Following the previous study [1], we define three question types, *first*, *no-switch*, and *switch*. The *first* question is literally first question of conversation without any history. The *no-switch* and *switch* questions can be distinguished by whether $d_{t}$ contains similar or same topics as $d_{t+1}$, where the $d_{t}$ is a gold passage at turn $t$ and $t > 1$. We conduct experiments on the topic-switching dataset TopiOCQA to analysis how the quality of LLM-generated supervision signals varies across different types of questions. As shown in **Table 1**, we find that **powerful ad-hoc retrievers like Contriever can solve the *first* problem very well, but can hardly handle the *switch* question**. **After the guidance of our InstructoR, Contriever achieves the most significant performance improvement for the *switch* problem**, indicating that LLM-generated supervision signals can help the retriever understand the complex conversation session and discover the user’s query intent in topic-switching scenarios. We think your suggestion is well, and we will add it in the revised version.
>
> **Table 1:** The performance of different question types (All, First, No-switch and Switch).
>
> | Retriever          | MRR (All) | NDCG@3 (All) | R@10 (All) | R@100 (All) | MRR (First) | NDCG@3 (First) | R@10 (First) | R@100 (First) | MRR (No-switch) | NDCG@3 (No-switch) | R@10 (No-switch) | R@100 (No-switch) | MRR (Switch) | NDCG@3 (Switch) | R@10 (Switch) | R@100 (Switch) |
> | ------------------ | :-------: | :----------: | :--------: | :---------: | :---------: | :------------: | :----------: | :-----------: | :-------------: | :----------------: | :--------------: | :---------------: | :----------: | :-------------: | :-----------: | :------------: |
> | Contriever         |    6.9    |     5.3      |    14.5    |    37.5     |     8.4     |      5.1       |     23.4     |     60.5      |       8.6       |        6.8         |       17.9       |       42.9        |     2.2      |       1.6       |      3.6      |      17.1      |
> | &nbsp;&nbsp;&nbsp;w/ InstructoR CRCG |   16.4    |     13.7     |    34.0    |    67.5     |    17.7     |      14.7      |     39.5     |     80.5      |      19.3       |        16.8        |       38.5       |       71.2        |     8.8      |       5.8       |     21.6      |      54.2      |
> | &nbsp;&nbsp;&nbsp;w/ InstructoR QRLV |   18.6    |     15.9     |    37.7    |    71.8     |    19.0     |      15.9      |     43.4     |     81.0      |      21.7       |        18.8        |       42.6       |       75.0        |     11.2     |       8.7       |     24.0      |      61.0      |
> | &nbsp;&nbsp;&nbsp;w/ InstructoR QRPG |   27.4    |     25.2     |    51.3    |    78.1     |    27.2     |      25.7      |     55.6     |     85.4      |      27.7       |        25.2        |       52.8       |       79.1        |     20.4     |      17.9       |     41.4      |      71.8      |
> | Contriever-msmarco |   16.2    |     14.3     |    29.4    |    56.0     |    36.2     |      33.2      |     61.0     |     **89.3**      |      17.9       |        15.7        |       32.3       |       60.7        |     6.1      |       4.9       |     12.4      |      34.1      |
> | &nbsp;&nbsp;&nbsp;w/ InstructoR CRCG |   23.9    |     21.6     |    44.4    |    73.7     |    38.3     |      35.5      |     **66.3**     |     **89.3**      |      27.0       |        25.1        |       47.7       |       75.1        |     15.5     |      13.7       |     28.9      |      60.5      |
> | &nbsp;&nbsp;&nbsp;w/ InstructoR QRLV |   28.6    |     26.0     |    52.3    |    80.2     |    34.8     |      33.2      |     63.9     |     84.9      |      31.1       |        28.5        |       56.2       |       84.1        |     19.2     |      16.9       |     36.7      |      65.9      |
> | &nbsp;&nbsp;&nbsp;w/ InstructoR QRPG |   **38.5**    |     **37.0**     |    **62.1**    |    **83.2**     |    **46.4**     |      **45.5**      |     65.9     |     87.8      |      **39.5**       |        **37.7**        |       **64.0**       |       **84.6**        |     **33.5**     |      **32.5**       |     **56.0**      |      **78.4**      |
>
> [1] Saving Dense Retriever from Shortcut Dependency in Conversational Search, Kim et al. 2022.
>
> > **Response to Rejection 2**: Unclear how the approach scales as the dataset size increases. Memory and compute costs?
>
> Thanks for your valuable comments! We will add the following discussion in the Appendix.
>
> Our proposed framework is flexible and scalable, and we analyze it from memory and compute costs.
>
> - Memory cost: The main memory cost of our method consists of three components: the passage index, the retriever, and the LLM. We compute the memory cost as follows: $C_{memory}=d\times|D|+2\times\theta_{retriever}+\theta_{LLM}$, where $d=768$ is the dimension of the session/passage embeddings, $|D|$ is the total number of passages,  $\theta_{retriever}=110M$ is the parameter size of the retriever, and $\theta_{LLM}=\{80M,250M,780M,3B\}$ is the parameter size of the LLM. As with most retrieval methods, our memory cost is mainly in $d\times|D|$. We take Flan-T5 XL ($\theta_{LLM}=3B$ ) as an example and provide memory consumption on QReCC and TopiOCQA. As shown in **Table 2**, our method is friendly to most university labs, which can easily run on a single machine with 2-4 GPUs.
>
> **Table 2:** The memory cost on QReCC and TopiOCQA.
> |             | **QReCC** | **TopiOCQA** |
> | :---------- | :-------: | :----------: |
> | Memory Cost |  44.69B   |    22.42B    |
> | GPU Cost    |  4*A6000  |   2*A6000    |
>
> - Compute cost: The main compute cost of our method is concentrated at the training stage. We need to emphasize that our method only needs to use LLMs during the training stage, so InstructoR is computationally efficient during the inference phase. We use the FLOPs-per-token estimates to calculate the compute cost: $C_{compute}=l_{session}\times\theta_{retriever}+|D|\times(d+d-1)+K\times(l_{session}+l_{passage})\times\theta_{LLM}$, where $l_{session}=200$ is the number of tokens per session, $l_{passage}=100$ is the number of tokens per passage, and $K=\{8,16,32\}$ is the number of retrieved top-K passages. We provide the specific compute cost of our method in **Table 3**. In our implementation, our method requires only a few hours for training. We also report the cost of calling ChatGPT API in our paper (line 480), which only costs $5 to generate 10,000 query rewrites.
>
> **Table 3:** The compute cost on QReCC and TopiOCQA.
> |                  | **QReCC** | **TopiOCQA** |
> | ---------------- | :-------: | :----------: |
> | FLOPs Per Sample | 2.890*e13 |  2.886*e13   |
> | Compute Cost     | 1.835*e18 |  1.311*e18   |
>
> To further reduce the memory and compute costs, we can use cheaper inference (e.g., LLM.fp16 and LLM.int8) and smaller LLM (e.g., Flan-T5 base and Flan-T5 large). We also examine the effect of LLM sizes on retrieval performance improvements in Appendix F. We find that using a small language model such as Flan-T5 base (250M) can also achieve good results.
>
> **We then discuss the latency of our method**. Our method only needs to distill knowledge from LLM into the retriever at the training stage, and does not need to use LLM at the inference stage. Hence, **our method is efficient in inference and has the same low latency as other dense retrieval methods**.
>
> > **Response to Question 1**: Can you provide more analysis into how the quality of LLM-generated supervision signals varies across different types of conversations or questions? Any patterns in where it works better/worse?
>
> Please refer to **Response to Rejection 1**.
>
> > **Writing Comments**: Typos Grammar Style And Presentation Improvements
>
> We really appreciate your suggestions on the details of our paper writing! Based on your suggestions, we now revise typos and use sub-section headings for each strategy.

---

### Official Review · Reviewer_nZaB · 2023-08-03

**Soundness:** 3

**Excitement:**

3: Ambivalent: It has merits (e.g., it reports state-of-the-art results, the idea is nice), but there are key weaknesses (e.g., it describes incremental work), and it can significantly benefit from another round of revision. However, I won't object to accepting it if my co-reviewers champion it.

**Missing References:**

[1] Large Language Models Know Your Contextual Search Intent: A Prompting Framework for Conversational Search, Mao et al., 2023

[2] ConvGQR: Generative Query Reformulation for Conversational Search, Mo et al., 2023

[3] Learning to Relate to Previous Turns in Conversational Search, Mo et al., 2023


**Paper Topic And Main Contributions:**

This paper proposes a new unsupervised method “Instructor” for training conversational dense retrieval models. Instructor leverages the knowledge from large language models (LLMs) to generate supervision signals to instruct the retriever. To distill the knowledge from the LLMs to the retriever, the authors design three methods to calculate the relevance score more precisely. Sufficient experimental results on four datasets under various settings demonstrate the effectiveness of the proposed method. Overall, the paper is well-written and easy to understand. The experiments look very solid. Though the technical depth is not so deep, and the distillation idea is not very novel, I suggest a weak accept on this paper.

**Questions For The Authors:**

Q1: How do you compute the confidence $\log p(r|I^{r}_{c,q},c,q)$ by ChatGPT? It seems that it cannot return the generation probability through API. Even if it is text-davinci, it can only provide the probability of the first five tokens. How do you deal with this problem?

Q2: The proposed method is applied to several ad-hoc dense retriever. Is it also possible to apply it for conversational dense retrieval model?


**Reasons To Accept:**

1. The proposed method leverages LLMs for knowledge distillation, which is a good application of LLMs on real research.
2. The experiments are sufficient and solid. The results show significant improvement over existing studies.
3. The paper is well-written and easy to understand.


**Reasons To Reject:**

1. Distilling knowledge from LLM to retriever, or more generally, distilling knowledge from reranker to retriever is not a new idea [1, 2].
2. Missing some recent literature in the related work part.

Reference:

1. Is ChatGPT Good at Search? Investigating Large Language Models as Re-Ranking Agent, Sun et al. 2023

2. Large language models are effective text rankers with pairwise ranking prompting, Qin et al., 2023


**Reproducibility:**

4: Could mostly reproduce the results, but there may be some variation because of sample variance or minor variations in their interpretation of the protocol or method.

**Reviewer Confidence:**

4: Quite sure. I tried to check the important points carefully. It's unlikely, though conceivable, that I missed something that should affect my ratings.

---

> ### Author Rebuttal · Authors · 2023-08-29
>
> Thanks for your careful and insightful reviews. Your professional reviews offer us great advice towards writing a more comprehensive and competitive paper!
>
> > **Response to Rejection 1**: Distilling knowledge from LLM to retriever, or more generally, distilling knowledge from reranker to retriever is not a new idea [1, 2].
>
> **We first clarify the differences and advantages between our InstructoR and these methods that also use LLMs for retrieval**.
>
> - Both RankGPT [1] and PRP [2] are very inspiring work, which use LLM to rank top-K passages at the reranking stage. However, it is difficult to handle millions of passages at the retrieval stage. Compared with these methods, **our method is more efficient with less latency**. Because our method only needs to distill knowledge from LLM into the retriever at the training stage, and does not need to use LLM at the inference stage.
> - Besides, there are some interesting work [3, 4] that leverage LLM as dataset generator, and train task-specific retrievers based on the generated data. Different from these outstanding work, our method does not need to use LLM to generate training data, but unleashes the power of LLM to judge the relevance between sessions and passages. **We try to avoid the bias and noise brought by the generated text as much as possible**.
> - To the best of our knowledge, **this is the first attempt to utilize LLMs to improve conversational dense retrieval in an unsupervised manner**. Specially, we devise three instructing strategies from context, query and response perspectives in the conversation. All three strategies are tailored for conversational dense retrieval. Furthermore, our method provides a feasible idea to combine white-box LLMs and black-box LLMs for distilling knowledge from LLM to retriever.
>
> [1] Is ChatGPT Good at Search? Investigating Large Language Models as Re-Ranking Agent, Sun et al. 2023.
>
> [2] Large language models are effective text rankers with pairwise ranking prompting, Qin et al., 2023.
>
> [3] Promptagator: Few-shot Dense Retrieval From 8 Examples, Dai et al. 2022.
>
> [4] InPars: Data Augmentation for Information Retrieval using Large Language Models, Luiz et al., 2022.
>
> > **Response to Rejection 2**: Missing some recent literature in the related work part.
>
> We really appreciate your suggestions on the details of our paper writing! Based on your suggestions, we now add the discussions of missing references [5, 6, 7] in the related work section. We also compare our method with the latest query reformulation methods (e.g., ConvGQR) in the experiment section.
>
> [5] Large Language Models Know Your Contextual Search Intent: A Prompting Framework for Conversational Search, Mao et al., 2023.
>
> [6] ConvGQR: Generative Query Reformulation for Conversational Search, Mo et al., 2023.
>
> [7] Learning to Relate to Previous Turns in Conversational Search, Mo et al., 2023.
>
> > **Response to Question 1**: How do you compute the confidence $\log p(r\mid I_{c,q}^{r},c,q)$ by ChatGPT? It seems that it cannot return the generation probability through API. Even if it is text-davinci, it can only provide the probability of the first five tokens. How do you deal with this problem?
>
> We are sorry for making the confusions. In practical implementation, we first prompt ChatGPT to generate the query rewrites, then use Flan-T5 to compute the confidence $\log p(r\mid I_{c,q}^{r},c,q)$. We find that what plays a key role in Eq. 8 is $\log p(r \mid I_{z}^{r}, z)$.
>
> > **Response to Question 2**: The proposed method is applied to several ad-hoc dense retriever. Is it also possible to apply it for conversational dense retrieval model?
>
> Yes, **our method can also be applied to conversational dense retrievers**. **To be consistent with previous work and achieve a fair comparison, we use the widely used ad-hoc retrievers as the initialization**. Experimental results show InstructoR can bring significant improvements across various ad-hoc retrievers, even surpassing the supervised state-of-the-art conversational dense retrieval method. In addition, we consider the more realistic setting where session-passage pairs are not available, so we train an ad-hoc retriever in an unsupervised way to obtain a conversational retriever.
>
> > **Missing References**
>
> We really appreciate your suggestions on the details of our paper writing! Based on your suggestions, we now add the discussions of missing references [1, 2, 3, 4, 5, 6, 7] in the related work section. We also compare our method with the latest query reformulation methods in the experiment section.

---

### Official Review · Reviewer_UCrL · 2023-08-10

**Soundness:** 3

**Excitement:**

3: Ambivalent: It has merits (e.g., it reports state-of-the-art results, the idea is nice), but there are key weaknesses (e.g., it describes incremental work), and it can significantly benefit from another round of revision. However, I won't object to accepting it if my co-reviewers champion it.

**Missing References:**

1. Open-retrieval conversational question answering. Chen Qu, Liu Yang, Cen Chen, Minghui Qiu, W Bruce Croft, Mohit Iyyer. (SIGIR 2020)

2. Explicit Query Rewriting for Conversational Dense Retrieval. Hongjin Qian, Zhicheng Dou. (EMNLP 2022)

3. ConvGQR: Generative Query Reformulation for Conversational Search. Fengran Mo, Kelong Mao, Yutao Zhu, Yihong Wu, Kaiyu Huang, Jian-Yun Nie (ACL 2023)

4. Learning to Relate to Previous Turns in Conversational Search. Fengran Mo, Jian-Yun Nie, Kaiyu Huang, Kelong Mao, Yutao Zhu, Peng Li, Yang Liu (SIGKDD 2023)

5. CoSPLADE: Contextualizing SPLADE for Conversational Information Retrieval. Nam Hai Le, Thomas Gerald, Thibault Formal, Jian-Yun Nie, Benjamin Piwowarski, Laure Soulier. (ECIR 2023)

The query reformulation methods could be included to be discussed and some recent publications could be added to the related work later to make the literature review more complete.

**Paper Topic And Main Contributions:**

This paper proposes a new method called INSTRUCTOR for training conversational dense retrievers in an unsupervised manner, without needing labeled session-passage pairs. The key problem it addresses is the lack of training data for conversational retrieval, where annotating session-passage relevance is difficult. Most existing methods rely on supervised fine-tuning.

The main contributions are:

1. An unsupervised training framework where large language models estimate session-passage relevance scores to guide retriever training.

2. Three strategies to more accurately calculate relevance using the language model: modeling retrieval as conversation generation, question rewriting as a latent variable, and using question responses as a posterior guide.

3. Experiments showing INSTRUCTOR significantly improves various ad-hoc retrievers like DPR and ANCE, even surpassing supervised methods.

In summary, the paper proposes a novel unsupervised approach for training conversational retrievers by instructing them with large language models. The method and thorough experiments are the key contributions.

**Questions For The Authors:**

1. Can this framework designed without LLM? (i.e. How should we think is this method flexible/feasible or not) and how can we ensure the reproduction as the current LLM normally with fast version iteration. If not, is it still a flexible framework?

2. What is the cost and the latency of using LLM, especially on the two big QReCC and TopiOCQA datasets? Please show statistic information.

3. How can we control the generated text by LLM will not bring noise to the query encoder training and how can we analysis via experiments? In other words, how to evaluate the quality of the generated text by LLM for IR evaluation.

4. How to consider the comparison with the manually rewritten query and the query reformulation approaches, especially on the cast datasets.

5. For Eq. 7, why we can assume log p(z), log p(c) and log p(q | c) as instants and what is the relation between eq. 7 and the quality of the generated text?

**Reasons To Accept:**

1. Addresses an important practical problem that lacking of conversational retrieval training data and proposes a creative method for leveraging large language models to provide training signal without any data annotation.

2. Thorough and rigorous experiments across diverse settings like low-resource and zero-shot. Surpassing supervised methods is a notable result.

Overall, the novel unsupervised learning approach, thorough experiments, and analyses make this a valuable contribution.

**Reasons To Reject:**

1. The approach relies heavily on large proprietary language models, which have limitations like bias and lack of transparency and there are limited error analysis to understand cases where the method fails. This could be important since the LLM is used as a black-box model and the quality of generated results are without guarantee. Meanwhile, the workflow details (e.g. the concrete prompt and the reformulated query) for the proposed three strategies should be much clearer.

2. There is no detailed analysis on how the unsupervised training objectives affect the retriever representations and no comparison with query reformulation methods. Besides, the cost and the latency of using LLM should also be indicated.

3. Lack of control of injecting noise (bring by the generated text from LLM) and the analysis of how these expansion terms will influence the retrieval results. This is important in terms of the query analysis and avoid the harmful terms generation.

The main risks relate to overselling the generality of the approach and glossing over potential issues with language models. But the paper seems reasonably cautious about limitations. Overall the methodology appears solid despite some aspects needing deeper analysis.

**Reproducibility:**

3: Could reproduce the results with some difficulty. The settings of parameters are underspecified or subjectively determined; the training/evaluation data are not widely available.

**Reviewer Confidence:**

4: Quite sure. I tried to check the important points carefully. It's unlikely, though conceivable, that I missed something that should affect my ratings.

---

> ### Author Rebuttal · Authors · 2023-08-29
>
> Thanks for your careful and insightful reviews. Your professional reviews offer us great advice towards writing a more comprehensive and competitive paper!
>
> > **Response to Rejection 1**: The approach relies heavily on large proprietary language models, which have limitations like bias and lack of transparency and there are limited error analysis to understand cases where the method fails. This could be important since the LLM is used as a black-box model and the quality of generated results are without guarantee. Meanwhile, the workflow details (e.g. the concrete prompt and the reformulated query) for the proposed three strategies should be much clearer.
>
> We agree with your concerns in this regard. We think that **our paper is an early exploration of LLM in information retrieval, aiming to utilize LLMs to improve conversational dense retrieval in an unsupervised manner**.
>
> While current LLMs have some limitations like bias and lack of transparency, we believe that this will be gradually solved in the future. In our work, we are cautious about these limitations. **To make our method clearer and more transparent, we provide the workflow details of our method in Appendix**. As shown in Appendix E, we provide all the instruction templates/prompts used in our strategies for reproduction. As shown in Table 4, 5, 6, 7, 8, and 9, we provide several reformulated query examples to better understand where LLM successes and fails. We can find that Black-box LLM ChatGPT achieves human-comparable or even better query rewriting quality. Sometimes ChatGPT is too smart and will answer questions redundantly (e.g., Table 7) or refuse to rewrite uncertain questions (e.g., Table 9). For white-box LLMs, the effect of query rewriting continues to improve as the LLM parameters increase. However, T0pp may not have been sufficiently fine-tuned with instruction data, so it is difficult to follow our instructions to generate rewrites. It is worth mentioning that Vicuna which is an open-source chatbot trained by fine-tuning LLaMA achieves the closest effect to ChatGPT.
>
> Furthermore, our work, as the first stage of retrieval-augmented methods, aims to better retrieve supported passages based on the current conversations. Because the retrieved passages can reduce the hallucinations and bias of LLMs, **we think that our work is a step towards improvement in these areas**. We agree the controllable generation of reformulated queries is very important, we leave it in future work.
>
> > **Response to Rejection 2**: There is no detailed analysis on how the unsupervised training objectives affect the retriever representations and no comparison with query reformulation methods. Besides, the cost and the latency of using LLM should also be indicated.
>
> Thanks for your valuable suggestion! We believe that **our unsupervised training objectives will lead to better representations of questions, especially topic-switching questions**. To analysis how the unsupervised training objectives affect the retriever representations, we define three question types, *first*, *no-switch*, and *switch* following the previous study [1]. The *first* question is literally first question of conversation without any history. The *no-switch* and *switch* questions can be distinguished by whether $d_{t}$ contains similar or same topics as $d_{t+1}$, where the $d_{t}$ is a gold passage at turn $t$ and $t > 1$. We conduct experiments on the topic-switching dataset TopiOCQA to analysis how the quality of LLM-generated supervision signals varies across different types of questions. As shown in **Table 1**, we find that **powerful ad-hoc retrievers like Contriever can solve the *first* problem very well, but can hardly handle the *switch* question**. **After the guidance of our InstructoR, Contriever achieves the most significant performance improvement for the *switch* problem**, indicating that LLM-generated supervision signals can help the retriever understand the complex conversation session and discover the user’s query intent in topic-switching scenarios. We think your suggestion is well, and we will add it in the revised version.
>
> **Table 1:** The performance of different question types (All, First, No-switch and Switch).
>
> | Retriever                             | MRR (All) | NDCG@3 (All) | R@10 (All) | R@100 (All) | MRR (First) | NDCG@3 (First) | R@10 (First) | R@100 (First) | MRR (No-switch) | NDCG@3 (No-switch) | R@10 (No-switch) | R@100 (No-switch) | MRR (Switch) | NDCG@3 (Switch) | R@10 (Switch) | R@100 (Switch) |
> | ------------------------------------- | :-------: | :----------: | :--------: | :---------: | :---------: | :------------: | :----------: | :-----------: | :-------------: | :----------------: | :--------------: | :---------------: | :----------: | :-------------: | :-----------: | :------------: |
> | Contriever                            |    6.9    |     5.3      |    14.5    |    37.5     |     8.4     |      5.1       |     23.4     |     60.5      |       8.6       |        6.8         |       17.9       |       42.9        |     2.2      |       1.6       |      3.6      |      17.1      |
> | &nbsp;&nbsp;&nbsp;w/ InstructoR CRCG  |   16.4    |     13.7     |    34.0    |    67.5     |    17.7     |      14.7      |     39.5     |     80.5      |      19.3       |        16.8        |       38.5       |       71.2        |     8.8      |       5.8       |     21.6      |      54.2      |
> | &nbsp;&nbsp;&nbsp;w/ InstructoR QRLV  |   18.6    |     15.9     |    37.7    |    71.8     |    19.0     |      15.9      |     43.4     |     81.0      |      21.7       |        18.8        |       42.6       |       75.0        |     11.2     |       8.7       |     24.0      |      61.0      |
> | &nbsp;&nbsp;&nbsp;w/ InstructoR QRPG  |   27.4    |     25.2     |    51.3    |    78.1     |    27.2     |      25.7      |     55.6     |     85.4      |      27.7       |        25.2        |       52.8       |       79.1        |     20.4     |      17.9       |     41.4      |      71.8      |
> | Contriever-msmarco                    |   16.2    |     14.3     |    29.4    |    56.0     |    36.2     |      33.2      |     61.0     |     **89.3**      |      17.9       |        15.7        |       32.3       |       60.7        |     6.1      |       4.9       |     12.4      |      34.1      |
> | &nbsp;&nbsp;&nbsp;w/ InstructoR CRCG  |   23.9    |     21.6     |    44.4    |    73.7     |    38.3     |      35.5      |    **66.3**     |     **89.3**      |      27.0       |        25.1        |       47.7       |       75.1        |     15.5     |      13.7       |     28.9      |      60.5      |
> | &nbsp;&nbsp;&nbsp;w/ InstructorR QRLV |   28.6    |     26.0     |    52.3    |    80.2     |    34.8     |      33.2      |     63.9     |     84.9      |      31.1       |        28.5        |       56.2       |       84.1        |     19.2     |      16.9       |     36.7      |      65.9      |
> | &nbsp;&nbsp;&nbsp;w/ InstructoR QRPG  |   **38.5**    |     **37.0**     |    **62.1**    |    **83.2**     |    **46.4**     |      **45.5**      |     65.9     |     87.8      |      **39.5**       |       **37.7**        |       **64.0**       |       **84.6**        |     **33.5**     |      **32.5**       |     **56.0**      |      **78.4**      |
>
> Besides, we also add more comparison with latest query reformulation methods in **Response to Question 4**.
>
> According to your suggestion, we discuss the cost and the latency of using LLM in **Response to Question 2**.
>
> [1] Saving Dense Retriever from Shortcut Dependency in Conversational Search, Kim et al. 2022.
>
> > **Response to Rejection 3**: Lack of control of injecting noise (bring by the generated text from LLM) and the analysis of how these expansion terms will influence the retrieval results. This is important in terms of the query analysis and avoid the harmful terms generation.
>
> **We would like to first clarify that our method mostly uses LLM to measure the relevance between input and output, and does not really need to generate text**. We try to avoid the bias and noise brought by the generated text as much as possible. Instead, we use the more transparent LLM generation probability as the retrieval guide. We only use LLM to generate query rewrites as the training data in the first step of QRLV strategy. Besides, **to ensure rewriting quality, we use regular expressions to filter out those rewrites that follow our instructions incorrectly as mentioned in Appendix D**.
>
> > **Response to Question 1**: Can this framework designed without LLM? (i.e. How should we think is this method flexible/feasible or not) and how can we ensure the reproduction as the current LLM normally with fast version iteration. If not, is it still a flexible framework?
>
> Yes, **our proposed framework is flexible and scalable**. In Appendix F, we examine the effect of LLM sizes on retrieval performance improvements. We find that using a small language model such as Flan-T5 base (250M) can also achieve good results. In Section 5.4, we analysis the selection of different types of LLM and find that our method is robust, with various LLMs leading to consistent improvements.
>
> To ensure the reproduction, we adopt the open-source models (e.g., Flan-T5) as the white-box LLMs. Considering that ChatGPT has a fast version iteration, we will release our generated rewrites for further research. **We believe the value of our work is to demonstrate the potential of utilizing LLMs to improve conversational dense retrieval in an unsupervised manner**.
>
> > **Response to Question 2**: What is the cost and the latency of using LLM, especially on the two big QReCC and TopiOCQA datasets? Please show statistic information.
>
> Thanks for your valuable comments! **We first analyze the cost of our method from memory and compute costs**.
>
> - Memory cost: The main memory cost of our method consists of three components: the passage index, the retriever, and the LLM. We compute the memory cost as follows: $C_{memory}=d\times|D|+2\times\theta_{retriever}+\theta_{LLM}$, where $d=768$ is the dimension of the session/passage embeddings, $|D|$ is the total number of passages,  $\theta_{retriever}=110M$ is the parameter size of the retriever, and $\theta_{LLM}=\{80M,250M,780M,3B\}$ is the parameter size of the LLM. As with most retrieval methods, our memory cost is mainly in $d\times|D|$. We take Flan-T5 XL ($\theta_{LLM}=3B$ ) as an example and provide memory consumption on QReCC and TopiOCQA. As shown in **Table 2**, our method is friendly to most university labs, which **can easily run on a single machine with 2-4 GPUs**.
>
> **Table 2:** The memory cost on QReCC and TopiOCQA.
>
> |             | **QReCC** | **TopiOCQA** |
> | :---------- | :-------: | :----------: |
> | Memory Cost |  44.69B   |    22.42B    |
> | GPU Cost    |  4*A6000  |   2*A6000    |
>
> Compute cost: The main compute cost of our method is concentrated at the training stage. We need to emphasize that our method only needs to use LLMs during the training stage, so **InstructoR is computationally efficient during the inference phase**. We use the FLOPs-per-token estimates to calculate the compute cost: $C_{compute}=l_{session}\times\theta_{retriever}+|D|\times(d+d-1)+K\times(l_{session}+l_{passage})\times\theta_{LLM}$, where $l_{session}=200$ is the number of tokens per session, $l_{passage}=100$ is the number of tokens per passage, and $K=\{8,16,32\}$ is the number of retrieved top-K passages. We provide the specific compute cost of our method in **Table 3**. In our implementation, our method requires only a few hours for training. We also report the cost of calling ChatGPT API in our paper (line 480), which only costs $5 to generate 10,000 query rewrites.
>
> **Table 3:** The compute cost on QReCC and TopiOCQA.
>
> |                  | **QReCC** | **TopiOCQA** |
> | ---------------- | :-------: | :----------: |
> | FLOPs Per Sample | 2.890*e13 |  2.886*e13   |
> | Compute Cost     | 1.835*e18 |  1.311*e18   |
>
> To further reduce the memory and compute costs, we can use cheaper inference (e.g., LLM.fp16 and LLM.int8) and smaller LLM (e.g., Flan-T5 base and Flan-T5 large). We also examine the effect of LLM sizes on retrieval performance improvements in Appendix F. We find that using a small language model such as Flan-T5 base (250M) can also achieve good results.
>
> **We then discuss the latency of our method**. Our method only needs to distill knowledge from LLM into the retriever at the training stage, and does not need to use LLM at the inference stage. Hence, **our method is efficient in inference and has the same low latency as other dense retrieval methods**.
>
> > **Response to Question 3**: How can we control the generated text by LLM will not bring noise to the query encoder training and how can we analysis via experiments? In other words, how to evaluate the quality of the generated text by LLM for IR evaluation.
>
> Please refer to **Response to Rejection 3**.
>
> > **Response to Question 4**: How to consider the comparison with the manually rewritten query and the query reformulation approaches, especially on the cast datasets.
>
> Thanks for your advice! We add the comparison with the manually rewritten query and the query reformulation approaches on TopiOCQA and CAsT datasets. To evaluate the effectiveness of our method, we compare it with several latest query reformulation baselines, including:
>
> - T5QR [2]: A strong T5-based QR model.
> - CONQRR [3]: A reinforcement-learning framework for T5-based QR model.
> - ConvGQR [4]: The current SOTA query reformulation method which integrates query rewriting and query expansion toward generating more effective search queries through a new knowledge infusion mechanism.
> - LLM Rewrite: A zero-shot ChatGPT-based model.
> - Human-Rewritten: Using manually rewritten query for retrieval.
>
> Experimental results validate the effectiveness of all three proposed instructing strategies on all the four datasets. **Our method surpasses the current supervised state-of-the-art query reformulation method ConvGQR**. Furthermore, **our method can achieve comparable or even better performance than Human-Rewritten**.
>
> **Table 4:** The performance on QReCC.
>
> | **Method**                           | **MRR**  | **NDCG@3** | **R@10** | **R@100** |
> | ------------------------------------ | :------: | :--------: | :------: | :-------: |
> | T5QR + ANCE                          |   34.5   |    31.8    |   53.1   |   72.8    |
> | T5QR + BM25                          |   33.4   |    30.2    |   53.8   |   86.1    |
> | CONQRR + ANCE                        |   41.8   |     -      |   65.1   |   84.7    |
> | CONQRR + BM25                        |   38.3   |     -      |   60.1   |   88.9    |
> | ConvGQR + ANCE                       |   42.0   |    39.1    |   63.5   |   81.8    |
> | ConvGQR + BM25                       |   44.1   |    41.0    |   54.4   |   88.0    |
> | LLM Rewrite + Contriever             |   19.7   |    17.0    |   33.6   |   56.4    |
> | LLM Rewrite + Contriever-msmarco     |   33.8   |    30.9    |   52.5   |   70.4    |
> | Human-Rewritten + ANCE               |   38.4   |    35.6    |   58.6   |   78.1    |
> | Human-Rewritten + BM25               |   39.7   |    36.2    |   62.5   | **98.5**  |
> | ANCE                                 |   41.4   |    38.9    |   61.2   |   74.7    |
> | &nbsp;&nbsp;&nbsp;w/ InstructoR CRCG |   42.0   |    39.1    |   64.6   |   83.0    |
> | &nbsp;&nbsp;&nbsp;w/ InstructoR QRLV |   42.7   |    40.2    |   63.4   |   78.4    |
> | &nbsp;&nbsp;&nbsp;w/ InstructoR QRPG |   43.5   |    40.5    |   66.7   |   85.6    |
> | Contriever-msmarco                   |   47.6   |    45.0    |   70.5   |   88.9    |
> | &nbsp;&nbsp;&nbsp;w/ InstructoR CRCG |   50.0   |    47.2    |   74.9   |   92.1    |
> | &nbsp;&nbsp;&nbsp;w/ InstructoR QRLV |   44.1   |    40.7    |   70.5   |   92.0    |
> | &nbsp;&nbsp;&nbsp;w/ InstructoR QRPG | **51.9** |  **49.1**  | **76.4** | **92.4**  |
>
> **Table 5:** The performance on TopiOCQA.
>
> | **Method**                           | **MRR**  | **NDCG@3** | **R@10** | **R@100** |
> | ------------------------------------ | :------: | :--------: | :------: | :-------: |
> | T5QR + ANCE                          |   23.0   |    22.2    |   37.6   |   54.4    |
> | T5QR + BM25                          |   11.3   |    9.8     |   22.1   |   44.7    |
> | CONQRR + ANCE                        |    -     |     -      |    -     |     -     |
> | CONQRR + BM25                        |    -     |     -      |    -     |     -     |
> | ConvGQR + ANCE                       |   25.6   |    24.3    |   41.8   |   58.8    |
> | ConvGQR + BM25                       |   12.4   |    10.7    |   23.8   |   45.6    |
> | LLM Rewrite + Contriever             |   12.4   |    10.3    |   26.2   |   52.4    |
> | LLM Rewrite + Contriever-msmarco     |   30.0   |    28.3    |   51.3   |   72.3    |
> | Human-Rewritten + ANCE               |    -     |     -      |    -     |     -     |
> | Human-Rewritten + BM25               |    -     |     -      |    -     |     -     |
> | ANCE                                 |   11.6   |    10.2    |   21.8   |   40.1    |
> | &nbsp;&nbsp;&nbsp;w/ InstructoR CRCG |   18.2   |    16.8    |   33.3   |   56.4    |
> | &nbsp;&nbsp;&nbsp;w/ InstructoR QRLV |   17.9   |    15.8    |   34.8   |   59.7    |
> | &nbsp;&nbsp;&nbsp;w/ InstructoR QRPG |   25.3   |    23.7    |   45.1   |   69.0    |
> | Contriever-msmarco                   |   16.2   |    14.3    |   29.4   |   56.0    |
> | &nbsp;&nbsp;&nbsp;w/ InstructoR CRCG |   23.9   |    21.6    |   44.4   |   73.7    |
> | &nbsp;&nbsp;&nbsp;w/ InstructoR QRLV |   28.6   |    26.0    |   52.3   |   80.2    |
> | &nbsp;&nbsp;&nbsp;w/ InstructoR QRPG | **38.5** |  **37.0**  | **62.1** | **83.2**  |
>
> **Table 6:** The performance on CAsT-19.
>
> | **Method**                       | **MRR**  | **NDCG@3** | **R@10** | **R@100** |
> | -------------------------------- | :------: | :--------: | :------: | :-------: |
> | T5QR                             |   70.1   |    41.7    |    -     |     -     |
> | ConvGQR                          |   70.8   |    43.4    |    -     |     -     |
> | LLM Rewrite + Contriever         |   40.6   |    24.5    |   6.4    |   26.1    |
> | LLM Rewrite + Contriever-msmarco |   63.6   |    48.2    |   12.2   |   43.8    |
> | Human-Rewritten                  | **74.0** |    46.1    |    -     |     -     |
> | Contriever                       |   25.2   |    12.7    |   3.1    |   13.8    |
> | &nbsp;&nbsp;&nbsp;w/ InstructoR  |   57.7   |    39.8    |   9.0    |   33.9    |
> | Contriever-msmarco               |   63.6   |    46.5    |   10.8   |   41.3    |
> | &nbsp;&nbsp;&nbsp;w/ InstructoR  | **70.4** |  **55.1**  | **13.2** | **45.4**  |
>
> **Table 7:** The performance on CAsT-20.
>
> | **Method**                       | **MRR**  | **NDCG@3** | **R@10** | **R@100** |
> | -------------------------------- | :------: | :--------: | :------: | :-------: |
> | T5QR                             |   42.3   |    29.9    |    -     |     -     |
> | ConvGQR                          |   46.5   |    **33.1**    |    -     |     -     |
> | LLM Rewrite + Contriever         |   21.7   |    12.0    |   7.5    |   25.1    |
> | LLM Rewrite + Contriever-msmarco |   39.6   |    25.7    |   14.6   |   43.5    |
> | Human-Rewritten                  | **59.1** |  **42.2**  |    -     |     -     |
> | Contriever                       |   18.8   |    9.3     |   8.2    |   25.2    |
> | &nbsp;&nbsp;&nbsp;w/ InstructoR  |   36.9   |    22.5    |   14.0   |   41.5    |
> | Contriever-msmarco               |   27.3   |    15.9    |   12.4   |   39.4    |
> | &nbsp;&nbsp;&nbsp;w/ InstructoR  | **51.7** |  32.8  | **18.7** | **54.0**  |
>
> [2] Conversational Question Reformulation via Sequence-to-Sequence Architectures and Pretrained Language Models, Wu et al. 2020.
>
> [3] CONQRR: Conversational Query Rewriting for Retrieval with Reinforcement Learning, Wu et al. 2022.
>
> [4] ConvGQR: Generative Query Reformulation for Conversational Search, Mo et al. 2023.
>
> > **Response to Question 5**: For Eq. 7, why we can assume $\log p(z)$, $\log p(c)$ and $\log p(q \mid  c)$ as instants and what is the relation between Eq. 7 and the quality of the generated text?
>
> We are sorry for making the confusions. In this paper, we assume that the passage prior $p(z)$ is uniform for all $z \in \mathcal{Z}$, and the conversation context prior $p(c)$ is uniform for all $c \in \mathcal{C}$. Since $c$ and $q$ happen at the same time, $p(q\mid c)=1$. Hence, we assume $\log p(z)$, $\log p(c)$ and $\log p(q \mid  c)$ as constants. Except for $\log p(r\mid I_{c,q}^{r},c,q)$ in Eq. 8, we use black-box LLMs to generate rewrites. We usually use LLMs to measure the relevance between input and output, not really generate text.
>
> > **Missing References**: The query reformulation methods could be included to be discussed and some recent publications could be added to the related work later to make the literature review more complete.
>
> We really appreciate your suggestions on the details of our paper writing! Based on your suggestions, we now add the discussions of missing references [2, 3, 4, 5, 6, 7, 8] in the related work section. We also compare our method with the latest query reformulation methods in the experiment section.
>
> [5] Open-retrieval Conversational Question Answering, Qu et al. 2020.
>
> [6] Explicit Query Rewriting for Conversational Dense Retrieval, Qian et al. 2022.
>
> [7] Learning to Relate to Previous Turns in Conversational Search, Mo et al. 2023.
>
> [8] CoSPLADE: Contextualizing SPLADE for Conversational Information Retrieval, Le et al. 2023.

---

### Official Review · Reviewer_WnM3 · 2023-08-11

**Soundness:** 3

**Excitement:**

3: Ambivalent: It has merits (e.g., it reports state-of-the-art results, the idea is nice), but there are key weaknesses (e.g., it describes incremental work), and it can significantly benefit from another round of revision. However, I won't object to accepting it if my co-reviewers champion it.

**Paper Topic And Main Contributions:**

The paper addresses the problem of conversational passage retrieval. The authors propose a method, named Instructor, to use frozen LLMs to provide retriever training signals. Specifically, the relevance score between a passage and a conversational query (i.e., conversation history + current user question) is calculated as the generation probability of the passage by a LLM, conditioning on the conversational query. The authors propose three ways to calculate such generation probabilities. Then they train the retriever with the KL loss between LM generation probabilities and passage-query similarities from the retriever.

They conduct experiments on three conversational retrieval benchmarks and achiever significant improvements with such training guidance from LLMs.

**Reasons To Accept:**

1. The paper is well-written and easy to follow.
2. They propose a novel framework to fine-tune the retriever model to handle conversational queries.
3. They did extensive experiments to show improvements of their approach on three different benchmarks.

**Reasons To Reject:**

1. My major concern is that you might be able to achieve impressive results with weak retrieval supervision from the conversation data, without the need to query LLMs as instructors. The training data used, namely QReCC and TopiOCQA, comes with both question and answer turns. You can apply the same method used in CONQRR (Wu et al., 2022) to derive "gold" passages and fine-tune retrievers with those weak labels. My guess is that the performance can already be good enough, which may also explain why QRPG gives the highest score as it uses the next answer turns. Also, you'll need to use hard negatives in retriever training as that's widely acknowledged to boost performance.

2. I don't agree with claim that "annotating session-passage pairs is much more difficult than collecting conversation data". The datasets you use are collected by having annotators search for passages first before writing agent utterances. How could annotating passages more difficult than labeling conversation data?

3. You are using LLM generation probability to calculate the document relevance, which is then used as training signals for retrievers. Why don't you simply use LLM generation prob to rank passages? You may get even better results than fine-tuning the retriever.

**Reproducibility:**

4: Could mostly reproduce the results, but there may be some variation because of sample variance or minor variations in their interpretation of the protocol or method.

**Reviewer Confidence:**

4: Quite sure. I tried to check the important points carefully. It's unlikely, though conceivable, that I missed something that should affect my ratings.

---

> ### Author Rebuttal · Authors · 2023-08-29
>
> Thanks for your careful and insightful reviews. Your professional reviews offer us great advice towards writing a more comprehensive and competitive paper!
>
> > **Response to Rejection 1**: My major concern is that you might be able to achieve impressive results with weak retrieval supervision from the conversation data, without the need to query LLMs as instructors. The training data used, namely QReCC and TopiOCQA, comes with both question and answer turns. You can apply the same method used in CONQRR [1] to derive "gold" passages and fine-tune retrievers with those weak labels. My guess is that the performance can already be good enough, which may also explain why QRPG gives the highest score as it uses the next answer turns. Also, you'll need to use hard negatives in retriever training as that's widely acknowledged to boost performance.
>
> **We would like to clarify the differences between InstructoR and the heuristics-based matching method employed in CONQRR and highlight the advantages of our method**.
>
> To annotate pseudo-gold passages as the weak retrieval supervision, CONQRR heuristically maps the answer $a_{t}$ to one passage with the **highest span overlap score**, and then designates it as the "gold" passage $d_{t}^{*}$ for the question $q_{t}$. Since the answers may be hidden in the original passages, this heuristics-based matching method has been widely adopted in open-domain QA [2]. The main differences and advantages between our InstructoR and this heuristics-based matching method are as follows:
> - In our training framework, the supervised labels generated by LLMs are **soft** (0 - 1). However, the labels obtained by heuristics rule are **hard** (0 or 1). For conversational retrieval, the relevance between session and passage may not be just 0 or 1. So **our method adopts more fine-grained labels**, achieving significant improvements across various ad-hoc retrievers. **It is worth mentioning that our method achieves even better performance than using real gold labels, which cannot be achieved using pseudo-gold labels**.
> - Our method employs LLMs to read sessions and passages and estimate the relevance scores, **which can better understand the relevances between sessions and passages than rule-based matching**. For example, it is difficult to find positive passages for boolean and abstractive answers using only string matching functions [3]. Retrievers can be affected by the noise introduced by the heuristics-based matching method.
> - While QRPG is only one of the three strategies we propose, **CRCG and QRLV strategies can still work well without using additional question responses**. Our method considers a more challenging unsupervised scenario, and provides effective solutions. **We also explore the use of LLM-generated responses in QRPG and demonstrate its effectiveness in Appendix I**.
> - Our method iteratively retrieves top-K passages and then scores them with LLM during the training stage. In contrast, the heuristics-based matching method involves dataset preprocessing before the training stage. **The retrieved top-K passages used for training can be considered as the hard samples [4] you mentioned, thus resulting in an improvement in our performance**.
>
> [1] CONQRR: Conversational Query Rewriting for Retrieval with Reinforcement Learning, Wu et al. 2022.
>
> [2] Dense Passage Retrieval for Open-Domain Question Answering, Karpukhin et al. 2020.
>
> [3] Neural Ranking with Weak Supervision for Open-Domain Question Answering: A Survey, Shen et al. 2023.
>
> [4] Approximate Nearest Neighbor Negative Contrastive Learning for Dense Text Retrieval, Xiong et al. 2021.
>
> > **Response to Rejection 2**: I don't agree with claim that "annotating session-passage pairs is much more difficult than collecting conversation data". The datasets you use are collected by having annotators search for passages first before writing agent utterances. How could annotating passages more difficult than labeling conversation data?
>
> We agree with your concerns in this regard. We want to explain why we claim that "annotating session-passage pairs is much more difficult than collecting conversation data".
>
> - We observe that a significant portion, specifically 53.4%, of the conversational questions present in the QReCC training set lacks corresponding labeled passages. Furthermore, it's worth noting that the CAsT datasets exclusively provide test sets without corresponding training sets. So we think **it is time-consuming and labor-intensive to label the source of the question responses**.
>
> - In practical applications, we can **collect rather than label** a large amount of public dialogue data through various sources (e.g., social media, chat platform and online forum), but the responses in these dialogues are often not clearly annotated with the sources (i.e., the supported passages).
> - In addition, some recent studies [5] have tried to use LLMs to automatically construct conversation data without human annotations, and **there will be no session-passage annotations for such synthetic conversations**. We think it is an interesting direction how to train a conversational retriever on such automatically synthesized conversation data. This could be a further extension of our work in the future.
>
> [5] q2d: Turning Questions into Dialogs to Teach Models How to Search, Bitton et al. 2023.
>
> > **Response to Rejection 3**: You are using LLM generation probability to calculate the document relevance, which is then used as training signals for retrievers. Why don't you simply use LLM generation prob to rank passages? You may get even better results than fine-tuning the retriever.
>
> Thanks for your valuable suggestion! We indeed explored the utilization of the LLM (Flan-T5 XL) generation probability to rerank the retrieved top-1000 passages. As shown in **Table 1**, we find that **the performance of LLM Rerank approach merely exhibits a marginal improvement compared to our CRCG strategy. However, it significantly lags behind the results achieved by both the QRLV and QRPG strategies**. We think this result is reasonable for the following reasons:
>
> - Our InstructoR adopts a trainable framework which iteratively mines top-K hard samples to **train the retriever in an unsupervised manner**. However, LLM rerank can only rank the passages **in a zero-shot manner**. Besides, using hard samples in the retriever training can boost performance.
> - During the training stage, QRLV models question rewrites generated by black-box LLMs as latent variables, and QRPG treats question responses as the posterior guide. **These useful supervised signals are not accessible to LLM Rerank method at the inference stage**.  So QRLV and QRPG strategies have achieved better performance than LLM Rerank.
> - LLM Rerank can only rank top-K (K<10000) retrieved passages at the reranking stage. However, it is difficult to handle **millions of passages** at the retrieval stage. Compared with LLM Rerank, **our method is more efficient with less latency**. Because our method only needs to distill knowledge from LLM into the retriever at the training stage, and does not need to use LLM at the inference stage.
>
> **Table 1:** The performance of LLM Rerank and InstructoR on TopiOCQA.
>
> | Retriever                            | MRR  | NDCG@3 | R@10 | R@100 |
> | ------------------------------------ | :--: | :----: | :--: | :---: |
> | Contriever                           | 6.9  |  5.3   | 14.5 | 37.5  |
> | &nbsp;&nbsp;&nbsp;w/ LLM Rerank      | 17.0 |  14.3  | 34.4 | 70.6  |
> | &nbsp;&nbsp;&nbsp;w/ InstructoR CRCG | 16.4 |  13.7  | 34.0 | 67.5  |
> | &nbsp;&nbsp;&nbsp;w/ InstructoR QRLV | 18.6 |  15.9  | 37.7 | 71.8  |
> | &nbsp;&nbsp;&nbsp;w/ InstructoR QRPG | 27.4 |  25.2  | 51.3 | 78.1  |
> | Contriever-msmarco                   | 16.2 |  14.3  | 29.4 | 56.0  |
> | &nbsp;&nbsp;&nbsp;w/ LLM Rerank      | 24.7 |  22.1  | 47.0 | 76.6  |
> | &nbsp;&nbsp;&nbsp;w/ InstructoR CRCG | 23.9 |  21.6  | 44.4 | 73.7  |
> | &nbsp;&nbsp;&nbsp;w/ InstructoR QRLV | 28.6 |  26.0  | 52.3 | 80.2  |
> | &nbsp;&nbsp;&nbsp;w/ InstructoR QRPG | **38.5** |  **37.0**  | **62.1** | **83.2**  |

---

### Meta-Review · Area_Chair_gwVm · 2023-09-19

**Recommendation:** 3

**Metareview:**

This paper proposes an alternative way to use LLMs for conversational IR. It generates relevance scores for candidate passages, and use the scores to instruct dense retrievers. Although there is some similarity between this method and the previous utilizations of LLM for generating supervision signals for dense retrievers, this study provides an alternative and interesting method.

The authors have answered most of the questions of the reviewers in their rebuttal. Overall, this is a valuable piece of work that enriches the current literature on conversational IR.

---

### Decision · Program_Chairs · 2023-10-07

**Decision:**

Accept-Findings

**Comment:**

This paper proposes an alternative way to use LLMs for conversational IR. It generates relevance scores for candidate passages, and use the scores to instruct dense retrievers. Although there is some similarity between this method and the previous utilizations of LLM for generating supervision signals for dense retrievers, this study provides an alternative and interesting method.

The authors have answered most of the questions of the reviewers in their rebuttal. Overall, this is a valuable piece of work that enriches the current literature on conversational IR.